# SYNAPSE: TRAJECTORY-AS-EXEMPLAR PROMPTING WITH MEMORY FOR COMPUTER CONTROL

**Longtao Zheng**[1] **Rundong Wang**[1] **Xinrun Wang**[1*] **Bo An**[1,2*]
[1]Nanyang Technological University, Singapore [2]Skywork AI, Singapore
https://ltzheng.github.io/Synapse

## ABSTRACT

Building agents with large language models (LLMs) for computer control is a burgeoning research area, where the agent receives computer states and performs actions to complete complex tasks. Previous computer agents have demonstrated the benefits of in-context learning (ICL); however, their performance is hindered by several issues. First, the limited context length of LLMs and complex computer states restrict the number of exemplars, as a single webpage can consume the entire context. Second, the exemplars in current methods, such as high-level plans and multi-choice questions, cannot represent complete trajectories, leading to suboptimal performance in long-horizon tasks. Third, existing computer agents rely on task-specific exemplars and overlook the similarity among tasks, resulting in poor generalization to novel tasks. To address these challenges, we introduce SYNAPSE, a computer agent featuring three key components: i) state abstraction, which filters out task-irrelevant information from raw states, allowing more exemplars within the limited context, ii) trajectory-as-exemplar prompting, which prompts the LLM with complete trajectories of the abstracted states and actions to improve multi-step decision-making, and iii) exemplar memory, which stores the embeddings of exemplars and retrieves them via similarity search for generalization to novel tasks. We evaluate SYNAPSE on MiniWoB++, a standard task suite, and Mind2Web, a real-world website benchmark. In MiniWoB++, SYNAPSE achieves a 99.2% average success rate (a 10% relative improvement) across 64 tasks using demonstrations from only 48 tasks. Notably, SYNAPSE is the first ICL method to solve the book-flight task in MiniWoB++. SYNAPSE also exhibits a 56% relative improvement in average step success rate over the previous state-of-the-art prompting scheme in Mind2Web.

## 1 INTRODUCTION

Creating agents capable of performing complex tasks to reduce human effort in routine operations is a long-standing goal in artificial intelligence (DeepMind Interactive Agents Team, 2021). A critical and pragmatic step towards this ambitious vision is to develop agents that are proficient in computer control (Shi et al., 2017; Liu et al., 2018). Such an agent perceives states of the computer, e.g., screenshots or webpage HTML, and performs actions via keyboard and mouse to complete tasks specified in natural language, such as flight booking and email management. Different from training-based methods (Humphreys et al., 2022) that rely on large-scale behavioral cloning (BC) (Pomerleau, 1989) and deep reinforcement learning (RL) (Sutton and Barto, 2018), recent studies highlight the advantages of in-context learning (ICL) (Brown et al., 2020; Wei et al., 2022a;b) in achieving general and data-efficient computer control (Yao et al., 2022a; Kim et al., 2023; Deng et al., 2023). Specifically, these computer agents utilize few-shot demonstrations, also termed exemplars, to prompt large language models (LLMs) (Brown et al., 2020; Chowdhery et al., 2022; OpenAI, 2023, *inter alia*) to generate actions. For example, RCI (Kim et al., 2023) prompts LLMs to generate a high-level plan and grounds each action based on the plan. Concurrent with our work, Deng et al. (2023) introduce Mind2Web as a benchmark for real-world web navigation and a computer agent, MindAct. It prompts LLMs to predict the next action using multi-choice questions (MCQ), with candidate choices generated by an element-ranking model.

---

*Corresponding authors

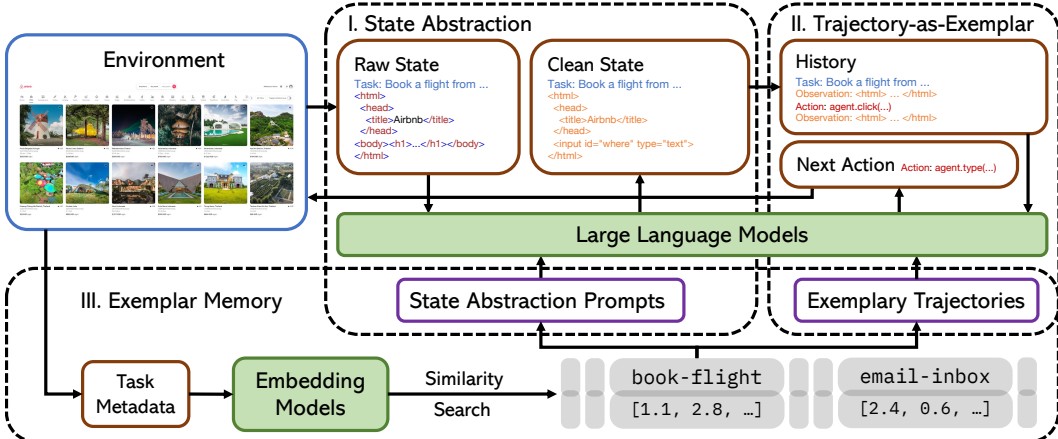

Figure 1: SYNAPSE consists of three main components. The process begins with **state abstraction**, where raw computer states (e.g., the HTML of webpages) are processed into concise task-relevant observations via few-shot learning of the LLM. This step reduces the number of tokens needed for each state, a prerequisite for the second component: **trajectory-as-exemplar (TaE) prompting**. In TaE prompting, the LLM is prompted with exemplary trajectories (a sequence of abstracted states and actions) and the current history to determine the next action. These prompts are retrieved from the **exemplar memory** using similarity search. The retrieval process utilizes the embeddings of task metadata, with each metadata mapped to the corresponding exemplars. Fig. 2 further shows an illustrative comparison of TaE prompting with the prompting schemes of other computer agents.

Despite the promising results of previous LLM-based computer agents, they still fail to solve some tasks within MiniWoB++ (Shi et al., 2017; Liu et al., 2018), a relatively simplified computer control task suite, mainly due to the following issues. First, the limited context length of LLMs, combined with complex computer states, poses challenges for few-shot learning. For example, the book-flight task in MiniWoB++ is particularly challenging due to the extensive length of the webpage displaying flight details. Also, the task-irrelevant information in raw states can distract LLMs from generating correct actions. Second, the exemplar structures used by existing computer agents can lead to error accumulation in multi-round LLM queries and struggle to capture the complete trajectory information. Specifically, existing computer agents query LLMs at each step, and neither high-level plans nor MCQs can represent full trajectories. For example, even when prompted with a correct step-by-step plan and previous actions, RCI may still incorrectly predict the next action. As a result, existing ICL methods struggle to solve long-horizon tasks. Even for simplified tasks, they still rely on self-correction. Third, current computer agents require task-specific exemplars within a predefined scope. For example, in MiniWoB++, both email-inbox and email-inbox-nl handle email-related tasks. Similarly, multi-orderings and multi-layouts share similar interfaces. However, existing methods overlook this task similarity and hard-code the mapping from tasks to exemplars.

To address these challenges, we present SYNAPSE, an LLM-powered computer agent featuring a novel prompting scheme and a memory mechanism. As illustrated in Fig. 1, SYNAPSE has three core components: i) state abstraction, which extracts task-relevant observations from raw states to reduce the token length per trajectory, ii) trajectory-as-exemplar (TaE) prompting, which utilizes full successful trajectories as few-shot exemplars to prompt the LLM for enhanced multi-step decision-making, and iii) exemplar memory, which stores successful trajectories and retrieves relevant ones via similarity search for generalization to novel tasks.

We thoroughly evaluate SYNAPSE on two benchmarks: MiniWoB++ (Shi et al., 2017; Liu et al., 2018), a standard research task suite, and Mind2Web (Deng et al., 2023), a dataset across diverse domains of real-world web navigation. In MiniWoB++, SYNAPSE is the first ICL method that achieves human-level performance (Fig. 3). Specifically, SYNAPSE solves 64 tasks with a success rate of 99.2% using demonstrations from only 48 tasks without relying on self-correction. In comparison, the current ICL state-of-the-art (SOTA) approach (Kim et al., 2023) is limited to handling 54 tasks, requiring task-specific demonstrations for each. Compared to BC+RL and fine-tuning SOTA baselines, SYNAPSE achieves superior performance while being simpler, more flexible, and

more sample-efficient. A notable achievement is its success on the book-flight task, which requires long-horizon decision-making under intricate states. We further demonstrate the effectiveness of our approach in Mind2Web by incrementally augmenting the LLM with three components of SYNAPSE, i.e., state abstraction, TaE prompting, and exemplar memory. When the underlying LLM is GPT-3.5, we achieve improvements of 32%, 50%, and 56% in the step success rate over MindAct, the SOTA ICL method in this domain. When we use CodeLlama-7B (Roziere et al., 2023), SYNAPSE also achieves an average 2.5× step success rate compared to MindAct. These experiments showcase the superiority of SYNAPSE over existing computer agents in both standard and real-world benchmarks.

## 2    RELATED WORK

**Building Agents with LLMs.**    There has been increasing attention on building autonomous agents with LLMs (Yang et al., 2023; Mialon et al., 2023; Xi et al., 2023; Park et al., 2023). Huang et al. (2022) and Brohan et al. (2023) combined the LLM-generated plan with an embedding and a value function to determine robot policies, respectively. Inner Monologue (Huang et al., 2023) incorporated environment feedback into a closed-loop system. These methods focused on prompting for better high-level semantic plans and relied on BC or RL for low-level motor control policies. Unlike robotics, low-level actions for computer control are highly semantic, and the effective prompting scheme for directly grounding actions remains unexplored. To address this, SYNAPSE proposes trajectory-as-exemplar prompting for directly grounding these actions. Recent efforts to improve the capabilities of reasoning, planning, and coding of LLMs, such as chain-of-thought (Wei et al., 2022a; Kojima et al., 2022), least-to-most (Zhou et al., 2022), ReAct (Yao et al., 2022a), tree-of-thoughts (Yao et al., 2023), Reflexion (Shinn et al., 2023), and self-debugging (Chen et al., 2023), can be combined with SYNAPSE to improve decision-making. The methods that leveraged external tools (Wu et al., 2023a; Shen et al., 2023; Schick et al., 2023; Lu et al., 2023; Xu et al., 2023) are also orthogonal to SYNAPSE. Code as Policies (Liang et al., 2023) and ProgPrompt (Singh et al., 2023) formulated agent policies as code generation. Similarly, SYNAPSE can be strengthened using code for both state abstraction and action generation. Voyager (Wang et al., 2023a) used a skill library for life-long skill acquisition, which also complements our framework. Wu et al. (2023b) also identified the issue of limited context, using a pretrained question-answering language model to filter states, similar to MindAct. In comparison, our state-abstraction principle can leverage both LLMs (few-shot learning) and pretrained state abstraction models.

**Agents for Computer Control.**    In the pursuit of creating agents capable of human-like computer interactions, MiniWoB++ task suite (Shi et al., 2017; Liu et al., 2018) has become a standard benchmark. Early attempts to solve MiniWoB++ primarily used BC and RL (Shi et al., 2017; Liu et al., 2018; Gur et al., 2018; Jia et al., 2018; Gur et al., 2021), which were not sufficient to achieve human-level performance. A notable breakthrough came with CC-Net (Humphreys et al., 2022), which achieved human-level performance but required an extensive dataset of 2.4 million demonstrations, equivalent to 6,300 hours of human effort. However, it is challenging to generalize this approach for new tasks and user customization. WebN-T5 (Gur et al., 2023a), a fine-tuned variant of the pretrained T5 model (Raffel et al., 2020), was trained on 12,000 demonstrations across 56 tasks. Similarly, WebGUM used 346,827 demonstrations to solve them. However, these fine-tuned LLMs still demanded a large amount of data and fell short of human performance in MiniWoB++. WebGPT (Nakano et al., 2021) and WebShop (Yao et al., 2022b), which operated in web-browsing environments, focused more on refining search queries rather than on general computer control. Recently, RCI (Kim et al., 2023) employed recursive self-correction to solve MiniWoB++ with a success rate of 90.6% in 54 tasks. However, it relied on task-specific exemplars, which limited its generalization to novel scenarios. In contrast, SYNAPSE achieves better performance in MiniWoB++ without relying on self-correction, using demonstrations from fewer tasks. There is also a lot of concurrent work in building computer agents and benchmarks. AdaPlanner (Sun et al., 2023) utilized environment feedback for self-correction and achieved a success rate of 92.9% in 53 tasks, but it had similar issues to RCI. Pix2Act (Shaw et al., 2023) solved 59 MiniWoB++ tasks with tree search and BC on 1.3 million demonstrations. WebAgent (Gur et al., 2023b) employed a small element-ranking model for HTML state filtering and an LLM for few-shot action generation, but it did not investigate the exemplar structure and memory. Among many sophisticated benchmarks, such as Mind2Web (Deng et al., 2023), AgentBench (Liu et al., 2023a), WebArena (Zhou et al., 2023), and Android in the Wild (Rawles et al., 2023), we choose Mind2Web for the real-world evaluation.

## 3 SYNAPSE

As illustrated in Fig. 1, SYNAPSE consists of three main components: state abstraction, trajectory-as-exemplar prompting, and exemplar memory. Here is the general pipeline of SYNAPSE: when a task occurs, SYNAPSE first retrieves relevant few-shot exemplars from memory by similarity search over the embedding space of task metadata, e.g., task descriptions and initial states. At each step, SYNAPSE first prompts the LLM to convert raw states into clean, task-relevant observations. Exemplary trajectories and the current trajectory (a task description and a sequence of observation-action pairs) are subsequently fed into the LLM to generate the next action.

### 3.1 PROBLEM SETTING

In this work, we focus on computer control tasks, where an agent interacts with the environment to accomplish a task specified by natural language. The state and action space for a computer agent are consistent with how humans interact with computers. At each step, it receives a computer state, such as HTML of webpages or screenshots, and performs actions via keyboard and mouse. The actions can be either code or natural language, depending on the benchmarks. As the decision-making engine, LLMs are fed with few-shot exemplars to process raw states into clean observations and generate actions. When the task is completed, the trajectory is considered successful.

### 3.2 STATE ABSTRACTION

The number of exemplars in few-shot learning can drastically influence the performance of LLM-powered computer agents (Wang et al., 2023b; Kim et al., 2023). However, the context limits of LLMs, combined with the intricacy of computer states, often restrict the number of exemplars. To illustrate, directly processing long HTML documents of real-world websites can be prohibitively costly because of many task-irrelevant elements. As a result, the performance of existing computer agents is hindered by the limited number of exemplars, and they often incorporate self-correction as a remedy. Moreover, many task-irrelevant elements in raw HTML states can potentially distract LLMs from accurate action generation. Even though some LLMs can handle longer inputs, their performance becomes worse as the input context grows longer (Liu et al., 2023b).

To reduce the length of each state, SYNAPSE takes advantage of the few-shot learning ability of LLMs to extract task-relevant information from raw states and form clean observations for subsequent action generation, as shown in Fig. 1. Specifically, we propose few-shot state abstraction prompts in both explicit and implicit forms. For scenarios where the context can handle multiple states, such as email-inbox in MiniWoB++, we apply explicit abstraction through state-observation pairs, denoted as ⟨state, observation⟩. In this case, the LLM is prompted with these pairs as few-shot exemplars, followed by the current raw state, to generate the clean observation. Alternatively, in scenarios with complex states where explicit abstraction is not applicable, such as book-flight in MiniWoB++, we adopt implicit abstraction by pairing task descriptions and state-parsing code, denoted as ⟨task, code⟩. We feed these pairs as few-shot exemplars to the LLM, along with the current task, to produce the code. This code takes the task and the raw state as parameters and returns the clean observation. If the code execution raises an error, we ask the LLM to perform zero-shot state abstraction. Within Mind2Web, we further demonstrate that the state abstraction principle can aid in effectively utilizing existing state abstraction tools. Specifically, Mind2Web has a pretrained element-ranking model that ranks and filters HTML elements to obtain a clean observation comprising the top-$k$ elements. As a simplified implementation of state abstraction, we set $k$ to 3 and 5 for the previous and current observations, respectively. In comparison, the SOTA method set $k$ to 50. Although this reduces the recall from 86% to 53%, i.e., only 53% of the clean observations contain the target element, it achieves a higher step success rate (Sec. 4.4). State abstraction shrinks long-winded raw states into concise and shorter ones, enabling trajectory-as-exemplar prompting (Sec. 3.3), which is impossible for previous methods due to context limits.

### 3.3 TRAJECTORY-AS-EXEMPLAR PROMPTING

The prompts for action generation in previous computer agents often overlook the sequential nature of decision-making. For example, each RCI exemplar consists of a high-level plan, while Mind-dAct formulates exemplars as step-by-step MCQs, as illustrated in Fig. 2. Without considering the

complete trajectories, these methods struggle to capture previous environment interactions. Furthermore, these methods query LLMs for one action at each step, leading to error accumulation over time. Therefore, these methods require self-correction even for simplified scenarios in MiniWoB++, especially for tasks with many steps and repeated actions, such as use-autocomplete and use-spinner.

Figure 2: Comparison of trajectory-as-exemplar prompting with other prompting schemes. The illustration is based on the terminal task in MiniWoB++, where the agent is asked to delete a file ending with a specific extension. To solve this task, RCI (Kim et al., 2023) prompts the LLM with a step-by-step plan combined with the current state and the previous actions to generate each action, while MindAct (Deng et al., 2023) prompts the LLM with MCQ-formatted exemplars at each step. In contrast, SYNAPSE uses a straightforward prompting scheme based on trajectory-level exemplars. This exemplar structure offers a consistent and interactive format, is more informative, and enables the LLM to produce temporally abstracted actions until a new state is required. As shown above, the LLM generates two consecutive actions: type(ls) and press(enter) without querying the new state. After executing these actions, it pauses to receive the new state for subsequent actions. We provide several complete trajectories in Appendix C for illustration.

To address this challenge, we introduce trajectory-as-exemplar (TaE) prompting, utilizing complete trajectories to prompt the LLM for action generation, formatted as ⟨task, observation, action, . . . , observation, action⟩. The first step of TaE prompting is to feed the LLM with successful trajectories. Subsequently, the LLM is prompted with the current trajectory (the task description and previous clean observations and actions) to produce the next action. Take Fig. 2 as an example, where the agent is asked to delete an item using the terminal. We pass a few successful trajectories to the LLM to illustrate how the problem is solved, followed by the current task description "Delete a file ending with the extension .gif" and the clean observation representing "the terminal is empty". We then execute the action returned by the LLM (agent.type(ls), agent.press(enter)) and construct the next prompt by appending the action and the next clean observation to the prompt. This process iterates until the agent completes the task or reaches the maximum number of steps. Similarly, in Mind2Web, we iteratively append states filtered by the element-ranking model and target actions to the prompt since it is a static dataset. This straightforward prompting scheme enhances the decision-making ability of the LLM for several reasons. First, the consistent format of interleaved observations and actions is well-suited for grounding actions, which allows conveniently parsing actions and setting stop tokens for LLM responses. Also, it provides more information for decision-making. More importantly, it implicitly prompts the LLM to generate temporally abstracted actions and query new states only when necessary. Temporal abstraction also results in lower cost and latency. Compared to human-crafted plans and MCQs, TaE exemplars can also be directly converted from human demonstrations. Therefore, TaE prompting improves the action accuracy in long-horizon tasks.

## 3.4 EXEMPLAR MEMORY

Previous computer agents rely on task-specific exemplars. For example, the mapping from tasks to few-shot exemplars is hard-coded in the RCI agent. Therefore, it requires exemplars for each of the 54 tasks tested, despite analogous tasks and interfaces, such as email-inbox and email-inbox-nl-turk, click-checkboxes-large and click-checkboxes-transfer, etc. In other words, existing computer agents lack a mechanism that matches relevant exemplars and tasks automatically. As a result, these agents cannot leverage task similarity and struggle to generalize to new tasks.

To exploit task similarity and enable generalization, SYNAPSE introduces exemplar memory $\mathcal{D} = (K, V)$, where $K$ is a fixed-sized array of embedding vectors of task metadata and $V$ is the cor-

responding state abstraction prompts and exemplary trajectories. This memory is constructed by encoding all the metadata using an embedding model and storing their corresponding exemplars in a vector database. As illustrated in Fig. 1, for a given task, SYNAPSE first encodes the task metadata and performs similarity search in the vector database, which retrieves relevant trajectories as few-shot exemplars. Formally, the retrieval process can be denoted as $\arg \text{top-}n_{d\in\mathcal{D}} \text{sim}(q, d)$ where $q$ is the query metadata and $\text{sim}$ is the Euclidean distance in the embedding space. In MiniWoB++, the metadata consists of the task description concatenated with the initial state from five random seeds of the selected 48 tasks. We retrieve the top three exemplars from memory and use the most common one to retrieve its exemplars, for there is a clear task division. For example, if the retrieved exemplars belong to enter-date, enter-date, and click-button, we retrieve all exemplars of enter-date. In Mind2Web, we encode the website name, domain, and task description as metadata for trajectories in the training set, and retrieve the corresponding exemplars of matched metadata for few-shot learning. Using the exemplar memory, SYNAPSE can automatically determine which exemplars to prompt, which lays the foundation for a general-purpose and adaptive computer agent.

# 4 EVALUATION

We evaluate our approach on two benchmarks, MiniWoB++ and Mind2Web. The MiniWoB++ task suite integrates diverse tasks that mirror the intricacies of real-world human-computer interactions (Shi et al., 2017; Liu et al., 2018). Although it is a relatively simplified environment, current computer agents struggle to solve several challenging tasks, such as book-flight, click-checkboxes-soft, and use-autocomplete. Mind2Web (Deng et al., 2023) is a realistic dataset containing human demonstrations of open-domain tasks from various real-world websites, such as Airbnb and Twitter. It measures in-the-wild generalization across different tasks, websites, and domains.

## 4.1 EXPERIMENTAL SETUP

To ensure fair comparisons, we query the same APIs of LLMs as in the prior work. In the MiniWoB++ experiments, we query `gpt-3.5-turbo-0301` and run 50 episodes to produce the results for each task. For Mind2Web, the default LLM is `gpt-3.5-turbo-16k-0613`. We configure the temperature to 0, i.e., greedy decoding. We use `text-embedding-ada-002` as the embedding model. For efficient retrieval in large-scale memory, we use Faiss (Johnson et al., 2019) to store embeddings and perform similarity search.

For the environment setup in MiniWoB++, we follow the configurations of RCI (Kim et al., 2023). The state space is the raw HTML code, while the action space includes click-xpath, move-mouse, type, press, and click-options. The main metric here is the success rate of the agent in accomplishing tasks. In Mind2Web, the observation space is the HTML provided in the dataset, and the action space includes click, type, and select. The metrics we measure include element selection accuracy (Ele. Acc), the success rate for each step in a task (Step SR), and the success rate for the whole task (SR) against the human-annotated ground truth. The Mind2Web dataset is split into a training set and three test sets: Cross-Task, Cross-Website, and Cross-Domain, to evaluate generalizability over tasks from the same websites, unseen websites from similar domains, and completely unseen domains in the training set, respectively. We only store the training set in the exemplar memory.

## 4.2 BASELINES

We conduct extensive experiments to evaluate the performance of SYNAPSE in comparison to the SOTA approaches on MiniWoB++. For BC+RL baselines, we use CC-Net (Humphreys et al., 2022) and Pix2Act (Shaw et al., 2023), both of which combine large-scale BC and RL. In terms of fine-tuning baselines, we compare with WebGUM (Furuta et al., 2023) and WebN-T5 (Gur et al., 2023a), two language models fine-tuned on a large number of demonstrations. As for ICL methods, the baselines are RCI (Kim et al., 2023) and AdaPlanner (Sun et al., 2023), both of which incorporate self-correction to solve 54 and 53 MiniWoB++ tasks, respectively. We also include human scores sourced from Humphreys et al. (2022) for additional benchmarking. In Mind2Web, we compare with MindAct (Deng et al., 2023), the current SOTA ICL method in this benchmark.

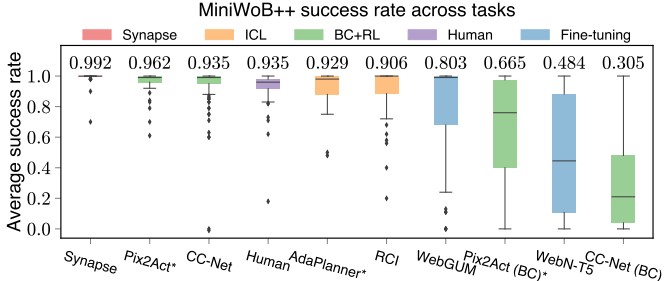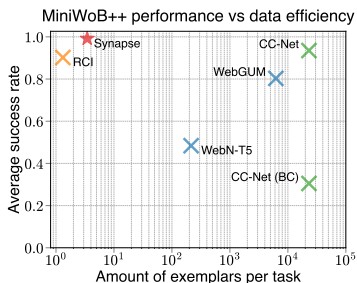

Figure 3: SYNAPSE is the first ICL method that achieves human-level performance in MiniWoB++. It outperforms previous self-correction methods, including RCI and AdaPlanner. A comprehensive task-wise evaluation is shown in Appendix A. *Pix2Act and AdaPlanner are concurrent with our work. We excluded them from the right-side figure due to their overlap with CC-Net and RCI. The outlier tasks are determined with an interquartile range of 1.5.

## 4.3 ANALYSIS ON MINIWOB++

Fig. 3 showcases the average performance of the various methods across tasks. With a mean success rate of 99.2%, SYNAPSE achieves human-level performance and outperforms all baselines on MiniWoB++. As shown in Fig. 4a, SYNAPSE outperforms BC+RL SOTA, especially on text-processing tasks, e.g., terminal and text-transform, while being more flexible. Compared to ICL (Fig. 4b) and fine-tuning SOTA (Fig. 4c), SYNAPSE achieves better performance on all tasks. It is particularly noteworthy that its superiority does not rely on self-correction. In comparison, RCI and AdaPlanner, although equipped with self-correction, fail to achieve comparable performance.

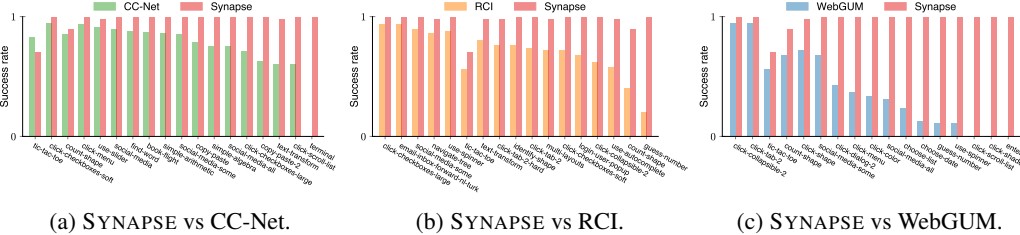

(a) SYNAPSE vs CC-Net.   (b) SYNAPSE vs RCI.   (c) SYNAPSE vs WebGUM.

Figure 4: Task-wise comparisons between SYNAPSE and various SOTA methods. For clarity, the bars in each figure are arranged in ascending order based on the success rate differences. Tasks not reported by other methods and those with differences less than 0.05 are not included.

Compared to previous ICL methods, SYNAPSE overcomes limitations related to context length, complicated states, and multi-step error accumulation (Fig. 4b). First, SYNAPSE outperforms them on tasks requiring detailed state understanding, such as click-collapsible-2, click-tab-2-hard, and count-shape. These results highlight the effectiveness of SYNAPSE in accurately capturing complex HTML states, while RCI consistently performs worse. Second, SYNAPSE significantly improves performance on tasks involving many steps or repeated actions by reducing the error accumulation in multi-round LLM queries, such as use-autocomplete, use-spinner, etc. Third, SYNAPSE can solve tasks that these approaches cannot handle due to the limited context length, such as book-flight and click-pie (Fig. 5). Moreover, it can generalize from existing exemplars to unseen tasks because it does not rely on task-specific exemplars. Therefore, SYNAPSE can solve 64 tasks, more than all previous ICL methods, using exemplars from only 48 tasks.

Most of the observed failure cases are due to incorrect reasoning by LLMs. For example, in some cases of the count-shape task, the LLM miscounts the number of target items, leading to the selection of an incorrect answer. Similarly, in the text-transform task, the LLM occasionally produces actions with characters similar to, but different from, those in the HTML (e.g., recognizing jrpf as jrfp).

## 4.4 EVALUATION ON REALISTIC WEBSITES

We also validate the effectiveness of the state abstraction principle, TaE prompting, and exemplar memory in the Mind2Web experiments. The current SOTA agent in this benchmark, MindAct, leverages an element-ranking model to filter the top-50 relevant elements as the clean observation. It uses MCQ to recursively query the LLM to select an action from five candidates until one action is chosen or all options are incorrect. Although MCQ-formatted exemplars outperform direct generation, MindAct often struggles to choose the correct element (Deng et al., 2023). In contrast, in our experiments, both SYNAPSE and its variants use direct generation, a more straightforward prompting scheme. As shown in Tab. 1, we incrementally equip direct generation with state abstraction, trajectory prompting, and exemplar memory, which achieves improvements of 32%, 50%, and 56% in average Step SR across three levels of generalization based on GPT-3.5, respectively. SYNAPSE also achieves 2.5× Step SR compared to MindAct on average based on CodeLlama-7B.

Table 1: Mind2Web results and ablations with CodeLlama-7B (top) and GPT-3.5 (bottom). We gradually add state abstraction, TaE prompting, and memory to demonstrate the effectiveness of each component. In SYNAPSE w/ state abstraction, we simply use direct generation with fewer top-ranked elements in clean observations, outperforming the MCQ-formatted prompting used in MindAct. SYNAPSE w/ state abstraction + TaE further shows the benefits of using trajectories as exemplars. In both these variants, we use static few-shot exemplars. Finally, we encode the training set as memory and retrieve exemplars via similarity search, which further improves performance.

| Method | Cross-Task | | | Cross-Website | | | Cross-Domain | | |
|---|---|---|---|---|---|---|---|---|---|
| | Ele. Acc | Step SR | SR | Ele. Acc | Step SR | SR | Ele. Acc | Step SR | SR |
| MindAct (CodeLlama-7B) | 11.2 | 7.7 | 0.4 | 12.4 | 9.0 | **0.6** | 13.8 | 9.9 | 0.2 |
| SYNAPSE w/ state abstraction | 21.4 | 15.5 | 0.8 | 19.5 | 13.5 | **0.6** | 20.0 | 15.6 | **1.2** |
| SYNAPSE w/ state abstraction + TaE | 27.0 | 24.5 | 1.2 | 21.4 | 17.9 | 0.0 | 22.1 | **19.9** | 0.7 |
| SYNAPSE w/ state abstraction + TaE + memory | **29.4** | **26.4** | **3.2** | **22.9** | **18.9** | **0.6** | **22.6** | 19.7 | 0.3 |
| MindAct (GPT-3.5) | 20.3 | 17.4 | 0.8 | 19.3 | 16.2 | **0.6** | 21.6 | 18.6 | 1.0 |
| SYNAPSE w/ state abstraction | 29.8 | 25.2 | 2.0 | 26.1 | 19.6 | **0.6** | 28.0 | 24.3 | **1.8** |
| SYNAPSE w/ state abstraction + TaE | 32.8 | 29.2 | 2.0 | 28.0 | 22.7 | **0.6** | 29.0 | 26.2 | **1.8** |
| SYNAPSE w/ state abstraction + TaE + memory | **34.0** | **30.6** | **2.4** | **29.1** | **24.2** | **0.6** | **29.6** | **26.4** | 1.5 |

## 4.5 ABLATION STUDIES

In this section, we perform ablation studies to evaluate the effectiveness of three design choices of SYNAPSE and provide insights into the contributions of each. These findings validate the effectiveness of i) state abstraction in handling complex states and providing more exemplars within the limited context, ii) TaE prompting in enhancing the capability of multi-step decision-making, and iii) exemplar memory in facilitating generalization across tasks.

**Ablating State Abstraction.** In Fig. 5, we illustrate the consistent performance improvement achieved by state abstraction in MiniWoB++. Due to the limited context, LLMs without state abstraction struggle to solve tasks with complex states, e.g., book-flight, while SYNAPSE effectively overcomes this. This technique also improves performance on tasks with ambiguous descriptions, such as email-inbox-nl-turk, increasing the success rate from 52% to 100%. In Mind2Web, direct generation with the top five elements improves the success rate by 32% compared to MCQ with the top-50 elements (Tab. 1), although reducing $k$ leads to a drop in recall from 86% to 53%. These results emphasize the effectiveness of state abstraction in helping LLMs generate actions under complex states and vague task descriptions.

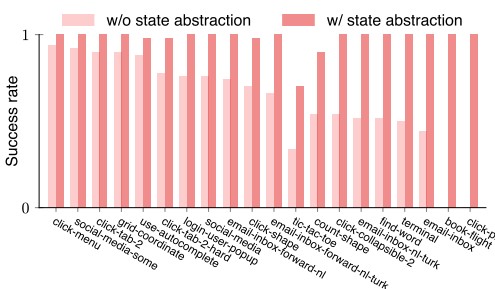

Figure 5: State abstraction enables solving MiniWoB++ tasks with complex states (e.g., book-flight). It also improves performance by providing more exemplars (e.g., email-inbox).

**Ablating Trajectory-as-Exemplar Prompting.** TaE prompting significantly benefits LLMs in tasks with long horizons and repeated actions such as guess-number, use-spinner, and use-autocomplete, which are particularly challenging for SOTA ICL methods. For example, RCI has success rates of 20%, 88%, and 58% for these tasks. In contrast, SYNAPSE solves them with success rates of 100%, 100%, and 98%. This improvement is attributed to the comprehensive history and temporal action abstraction introduced by TaE prompting. In Mind2Web, TaE prompting further improves the average step success rate by 19% compared to SYNAPSE with state abstraction only. It also outperforms MindAct by 50% across three levels of generalization, as shown in Tab. 1.

**Exemplar Memory Enables Generalization.** Exemplar memory is crucial for generalization, especially for similar computer control tasks. Rather than previous methods that rely on task-specific exemplars, SYNAPSE's memory can automatically find relevant exemplars for the task at hand. It enables the agent to identify and utilize exemplars to generalize from email-inbox-nl-turk to related tasks such as email-inbox-forward-nl-turk and email-inbox. It also adapts to different layouts, e.g., from

Table 2: The average distance of retrieval and Step SR improvements of memory ($\Delta$) in Mind2Web.

| Test set | Distance | $\Delta_{\text{GPT}}$ | $\Delta_{\text{CodeLlama}}$ |
|---|---|---|---|
| Cross-Task | 17.0 | 1.4 | 1.9 |
| Cross-Website | 24.3 | 1.5 | 1.0 |
| Cross-Domain | 32.9 | 0.2 | -0.2 |

multi-layouts to multi-orderings. Consequently, SYNAPSE solves 64 tasks in MiniWoB++ with demonstrations of only 48 tasks. It achieves better performance not only for the predefined tasks but also for new tasks outside the scope. For the 16 unseen tasks, the average success rate is almost 100%. For tasks within the scope, the memory mechanism also accurately matches relevant exemplars. SYNAPSE only encounters a 2% failure rate in click-tab-2-hard due to exemplar mismatches. In Mind2Web, the memory further improves SYNAPSE with state abstraction and TaE prompting, leading to a 6% improvement in Step SR for cross-task and cross-website generalization. However, the improvement is relatively marginal in cross-domain generalization, which might be attributed to the LLM being influenced by exemplars from unrelated domains. The average distance (similarity scores) between retrieved exemplars and the target task in three Mind2Web test sets are shown in Tab. 2. A greater distance implies less similarity. This accounts for why memory does not aid in cross-domain generalization, as these domains are entirely unseen.

## 5 DISCUSSION, LIMITATIONS & FUTURE WORK

In this paper, we introduce SYNAPSE, a computer agent with a prompting technique that is broadly applicable to any decision-making task. SYNAPSE addresses three main challenges faced by current computer agents: the limited context length, the unexplored exemplar structure, and task-specific exemplars. It has three key components: i) state abstraction, which converts raw states to clean observations, reducing the tokens of each state to allow for more few-shot exemplars, ii) trajectory-as-exemplar prompting, which prompts the LLM with trajectory-level exemplars to improve multi-step decision-making, and iii) exemplar memory, which stores and retrieves exemplars to enable generalization. Our method outperforms existing methods on 64 tasks in the MiniWoB++ benchmark with a 99.2% average success rate without self-correction. It is particularly noteworthy that our method successfully solves the challenging book-flight task in MiniWoB++. SYNAPSE also achieves a 56% relative improvement over SOTA ICL methods in Mind2Web, a real-world website benchmark, at three levels of generalization (cross-task, cross-website, and cross-domain).

We acknowledge that there exist limitations in our framework. High inference latency is a major concern due to the use of LLMs. Using our prompting scheme to distill a more responsive, task-specific agent from existing LLMs could be a remedy. Another concern is our dependence on the quality of exemplars. Therefore, training a zero-shot computer agent with instruction tuning (Wei et al., 2021; Chung et al., 2022) based on our prompting techniques is a promising research direction. Also, the memory module can be seamlessly combined with human intervention for user customization and task adaptation, which opens up an avenue for future research. In addition, the memory structure and retrieval process can be further investigated for better generalization. It is also worthwhile to combine our methods with compositional generalization methods (Zhou et al., 2022; Wang et al., 2023a) to solve more complicated tasks. Finally, while SYNAPSE currently operates on text, it would be interesting to explore multi-modal and video understanding capabilities to tackle more challenging tasks, e.g., pixel-based Android control (Li et al., 2020; Toyama et al., 2021).

ACKNOWLEDGEMENTS

We would like to thank the OpenAI Researcher Access Program for granting us API access for this project. This research is supported by the National Research Foundation Singapore and DSO National Laboratories under the AI Singapore Programme (AISGAward No: AISG2-GC-2023-009). This research is also supported by the National Research Foundation, Singapore under its Industry Alignment Fund – Pre-positioning (IAF-PP) Funding Initiative. Any opinions, findings and conclusions or recommendations expressed in this material are those of the author(s) and do not reflect the views of National Research Foundation, Singapore.

REPRODUCIBILITY STATEMENT

To ensure reproducibility, all resources such as code, prompts, and agent trajectories have been made publicly available at `https://ltzheng.github.io/Synapse`. Additionally, details on LLMs and environments are available in Sec. 4.1 and Appendix B.

ETHIC STATEMENT

Autonomous agents with computer control capabilities have inherent risks, such as performing dangerous operations and producing spam. To mitigate these risks, our experiments restrict agent interactions to controlled environments (MiniWoB++) and a static dataset (Mind2Web). Recognition and vigilance of these risks are crucial, especially when conducting real-world experiments or deploying these agents in practical settings.

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

## A  ADDITIONAL RESULTS

Table 3: Per-task mean success rate comparison between SYNAPSE, humans, and baselines. The data for human participants are obtained from CC-Net (Humphreys et al., 2022). The aggregated performance of the baselines prior to CC-Net is represented as others (Gur et al., 2018; Jia et al., 2018; Liu et al., 2018; Shi et al., 2017). We attach the average success rate and the number of solved tasks for each method at the bottom of the table.

| Task | Ours | Human | RCI | AdaPlanner | Pix2Act | Pix2Act (BC) | CC-Net | CC-Net (BC) | WebGUM | WebN-T5 | Others |
|---|---|---|---|---|---|---|---|---|---|---|---|
| bisect-angle | n/a | 0.92 | n/a | n/a | 0.96 | 0.32 | 0.97 | 0.29 | n/a | n/a | 0.80 |
| book-flight | 1.00 | 0.87 | n/a | n/a | n/a | n/a | 0.87 | 0.00 | 0.98 | 0.00 | 1.00 |
| chase-circle | n/a | 0.82 | n/a | n/a | n/a | n/a | 0.93 | 0.80 | n/a | n/a | 1.00 |
| choose-date | 1.00 | 0.97 | n/a | n/a | 0.79 | 0.06 | 0.97 | 0.12 | 0.13 | 0.00 | 1.00 |
| choose-date-easy | n/a | 0.99 | n/a | n/a | n/a | n/a | 0.99 | 0.42 | 1.00 | 0.03 | n/a |
| choose-date-medium | n/a | 0.98 | n/a | n/a | n/a | n/a | 0.99 | 0.26 | 0.60 | 0.00 | n/a |
| choose-list | 1.00 | 0.98 | 1.00 | 1.00 | n/a | n/a | 0.99 | 0.19 | 0.24 | 0.26 | 0.26 |
| circle-center | n/a | 0.96 | n/a | n/a | 0.96 | 0.52 | 0.97 | 0.36 | n/a | n/a | 0.98 |
| click-button | 1.00 | 0.98 | 1.00 | 1.00 | 0.99 | 0.32 | 1.00 | 0.78 | 1.00 | 1.00 | 1.00 |
| click-button-sequence | 1.00 | 0.94 | 1.00 | 1.00 | 0.99 | 1.00 | 1.00 | 0.47 | 1.00 | 1.00 | 1.00 |
| click-checkboxes | 1.00 | 0.97 | 1.00 | 1.00 | 1.00 | 0.99 | 0.98 | 0.32 | 1.00 | 0.96 | 1.00 |
| click-checkboxes-large | 1.00 | 0.87 | 0.94 | 1.00 | 0.99 | 1.00 | 0.71 | 0.00 | 0.99 | 0.22 | 0.84 |
| click-checkboxes-soft | 1.00 | 0.73 | 0.72 | 0.80 | 0.61 | 0.91 | 0.95 | 0.04 | 0.98 | 0.54 | 0.94 |
| click-checkboxes-transfer | 1.00 | 0.98 | 1.00 | 0.98 | 1.00 | 0.76 | 0.99 | 0.36 | 0.99 | 0.63 | 0.64 |
| click-collapsible | 1.00 | 0.99 | 1.00 | 1.00 | 0.94 | 0.80 | 1.00 | 0.81 | 0.98 | 0.00 | 1.00 |
| click-collapsible-2 | 1.00 | 0.97 | 0.62 | 0.84 | 0.97 | 0.31 | 0.98 | 0.17 | 0.95 | 0.00 | 0.99 |
| click-color | 1.00 | 0.97 | 1.00 | 1.00 | 0.99 | 0.88 | 1.00 | 0.82 | 0.34 | 0.27 | 1.00 |
| click-dialog | 1.00 | 1.00 | 1.00 | 1.00 | 1.00 | 0.12 | 1.00 | 0.95 | 1.00 | 1.00 | 1.00 |
| click-dialog-2 | 1.00 | 0.99 | 1.00 | 1.00 | 1.00 | 0.73 | 1.00 | 0.88 | 0.43 | 0.24 | 1.00 |
| click-link | 1.00 | 0.99 | 1.00 | 0.98 | 0.98 | 0.86 | 0.99 | 0.59 | 1.00 | 1.00 | 1.00 |
| click-menu | 1.00 | 0.97 | 1.00 | 0.78 | n/a | n/a | 0.94 | 0.22 | 0.37 | 0.37 | 0.13 |
| click-menu-2 | n/a | 0.98 | n/a | n/a | n/a | n/a | 0.83 | 0.52 | n/a | n/a | 0.16 |
| click-option | 1.00 | 0.99 | 1.00 | 1.00 | 1.00 | 0.00 | 0.99 | 0.21 | 1.00 | 0.87 | 1.00 |
| click-pie | 1.00 | 0.98 | n/a | n/a | 0.99 | 0.81 | 0.97 | 0.15 | 0.99 | 0.51 | 1.00 |
| click-scroll-list | 1.00 | 0.91 | 1.00 | 1.00 | n/a | n/a | 0.60 | 0.01 | 0.00 | 0.00 | 0.07 |
| click-shades | 1.00 | 0.91 | 1.00 | 1.00 | 0.99 | 0.76 | 1.00 | 0.04 | 0.00 | 0.00 | 0.99 |
| click-shape | 0.98 | 0.88 | 0.98 | 0.75 | 0.94 | 0.19 | 0.95 | 0.11 | 0.72 | 0.53 | 0.64 |
| click-tab | 1.00 | 0.99 | 1.00 | 1.00 | 1.00 | 0.54 | 1.00 | 0.95 | 1.00 | 0.74 | 1.00 |
| click-tab-2 | 1.00 | 0.97 | 0.74 | 0.85 | 0.98 | 0.42 | 0.98 | 0.27 | 0.95 | 0.18 | 1.00 |
| click-tab-2-easy | n/a | 0.99 | n/a | n/a | 0.99 | 0.77 | 0.99 | 0.61 | n/a | n/a | n/a |
| click-tab-2-hard | 0.98 | 0.96 | 0.76 | 0.78 | 0.97 | 0.00 | 0.98 | 0.19 | 0.95 | 0.12 | n/a |
| click-tab-2-medium | n/a | 0.97 | n/a | n/a | 1.00 | 0.07 | 0.99 | 0.54 | n/a | n/a | n/a |
| click-test | 1.00 | 1.00 | 1.00 | 1.00 | 1.00 | 1.00 | 1.00 | 1.00 | 1.00 | 1.00 | 1.00 |
| click-test-2 | 1.00 | 0.99 | 1.00 | 1.00 | 1.00 | 1.00 | 1.00 | 0.95 | 1.00 | 1.00 | 1.00 |
| click-test-transfer | n/a | 0.99 | n/a | n/a | 1.00 | 1.00 | 1.00 | 0.94 | n/a | n/a | n/a |
| click-widget | 1.00 | 0.83 | 0.98 | 1.00 | 1.00 | 0.87 | 1.00 | 0.56 | 1.00 | 1.00 | 1.00 |
| copy-paste | 1.00 | 0.94 | n/a | n/a | n/a | n/a | 0.79 | 0.04 | n/a | n/a | 0.00 |
| copy-paste-2 | 1.00 | 0.94 | n/a | n/a | n/a | n/a | 0.63 | 0.01 | n/a | n/a | 0.00 |
| count-shape | 0.90 | 0.82 | 0.40 | 0.50 | 0.70 | 0.00 | 0.85 | 0.21 | 0.68 | 0.41 | 0.76 |
| count-sides | n/a | 0.98 | n/a | n/a | 1.00 | 0.38 | 1.00 | 0.74 | n/a | n/a | 0.30 |
| drag-box | n/a | 0.99 | n/a | n/a | 0.99 | 1.00 | 1.00 | 0.61 | n/a | n/a | 0.31 |
| drag-cube | n/a | 0.99 | n/a | n/a | n/a | n/a | 0.79 | 0.23 | n/a | n/a | 0.18 |
| drag-item | n/a | 0.98 | n/a | n/a | 1.00 | 0.85 | 1.00 | 0.61 | n/a | n/a | n/a |
| drag-items | n/a | 0.93 | n/a | n/a | 1.00 | 0.64 | 0.99 | 0.13 | n/a | n/a | 0.41 |
| drag-items-grid | n/a | 0.87 | n/a | n/a | 0.89 | 0.60 | 0.98 | 0.05 | n/a | n/a | 0.01 |
| drag-shapes | n/a | 0.96 | n/a | n/a | 0.98 | 0.96 | 0.99 | 0.26 | n/a | n/a | 0.92 |
| drag-sort-numbers | n/a | 0.92 | n/a | n/a | 0.95 | 0.08 | 0.97 | 0.11 | n/a | n/a | 0.66 |
| email-inbox | 1.00 | 0.96 | 0.98 | 0.98 | n/a | n/a | 1.00 | 0.09 | 0.99 | 0.38 | 0.99 |
| email-inbox-delete | n/a | 0.99 | n/a | n/a | 1.00 | 0.99 | 1.00 | 0.22 | n/a | n/a | 1.00 |
| email-inbox-forward | n/a | 0.96 | n/a | n/a | n/a | n/a | 1.00 | 0.01 | n/a | n/a | n/a |
| email-inbox-forward-nl | 1.00 | 0.91 | 1.00 | 1.00 | n/a | n/a | 1.00 | 0.00 | 1.00 | 0.60 | n/a |
| email-inbox-forward-nl-turk | 1.00 | 0.88 | 0.94 | 1.00 | n/a | n/a | 1.00 | 0.00 | 1.00 | 0.33 | n/a |
| email-inbox-important | n/a | 0.99 | n/a | n/a | 1.00 | 0.99 | 1.00 | 0.30 | n/a | n/a | n/a |
| email-inbox-nl-turk | 1.00 | 0.93 | 0.98 | 0.90 | n/a | n/a | 1.00 | 0.05 | 0.98 | 0.23 | 0.93 |
| email-inbox-noscroll | n/a | 0.96 | n/a | n/a | n/a | n/a | 1.00 | 0.13 | n/a | n/a | n/a |
| email-inbox-reply | n/a | 0.91 | n/a | n/a | n/a | n/a | 1.00 | 0.00 | n/a | n/a | n/a |
| email-inbox-star-reply | n/a | 0.95 | n/a | n/a | n/a | n/a | 1.00 | 0.11 | n/a | n/a | n/a |
| enter-date | 1.00 | 0.97 | 0.96 | 1.00 | 1.00 | 0.59 | 1.00 | 0.02 | 1.00 | 0.00 | 1.00 |
| enter-password | 1.00 | 0.96 | 1.00 | 0.98 | n/a | n/a | 1.00 | 0.02 | 1.00 | 0.97 | 1.00 |
| enter-text | 1.00 | 0.98 | 1.00 | 0.98 | n/a | n/a | 1.00 | 0.35 | 1.00 | 0.89 | 1.00 |
| enter-text-2 | n/a | 0.91 | n/a | n/a | 0.97 | 1.00 | 0.98 | 0.04 | n/a | n/a | 0.00 |
| enter-text-dynamic | 1.00 | 0.97 | 1.00 | 0.96 | n/a | n/a | 1.00 | 0.39 | 1.00 | 0.98 | 1.00 |
| enter-time | 1.00 | 0.98 | 1.00 | 0.96 | 1.00 | 0.78 | 0.97 | 0.04 | 0.00 | 0.00 | 0.90 |
| find-midpoint | n/a | 0.94 | n/a | n/a | 0.96 | 0.74 | 0.97 | 0.35 | n/a | n/a | 0.31 |
| find-word | 1.00 | 0.96 | n/a | n/a | n/a | n/a | 0.88 | 0.05 | n/a | n/a | 0.00 |
| focus-text | 1.00 | 1.00 | 1.00 | 1.00 | n/a | n/a | 1.00 | 0.99 | 1.00 | 1.00 | 1.00 |
| focus-text-2 | 1.00 | 0.99 | 1.00 | 0.94 | n/a | n/a | 1.00 | 0.96 | 1.00 | 1.00 | 1.00 |
| grid-coordinate | 1.00 | 0.87 | 1.00 | 1.00 | 0.92 | 0.97 | 1.00 | 0.66 | 1.00 | 0.49 | 1.00 |
| guess-number | 1.00 | 0.99 | 0.20 | 0.88 | n/a | n/a | 1.00 | 0.21 | 0.11 | 0.00 | 0.20 |
| highlight-text | n/a | 0.97 | n/a | n/a | n/a | n/a | 1.00 | 0.51 | n/a | n/a | 0.90 |
| highlight-text-2 | n/a | 0.97 | n/a | n/a | n/a | n/a | 1.00 | 0.40 | n/a | n/a | 0.13 |
| identify-shape | 1.00 | 0.98 | 0.76 | 0.96 | 1.00 | 0.94 | 1.00 | 0.68 | 1.00 | 0.88 | 1.00 |
| login-user | 1.00 | 0.96 | 1.00 | 1.00 | n/a | n/a | 1.00 | 0.00 | 1.00 | 0.82 | 1.00 |
| login-user-popup | 1.00 | 0.94 | 0.68 | 0.98 | n/a | n/a | 1.00 | 0.02 | 0.99 | 0.72 | n/a |
| moving-items | n/a | 0.18 | n/a | n/a | n/a | n/a | 0.88 | 0.13 | n/a | n/a | 0.78 |
| multi-layouts | 0.98 | 0.95 | 0.72 | 0.84 | n/a | n/a | 1.00 | 0.00 | 1.00 | 0.83 | 1.00 |

| | | | | | | | | | | | |
|---|---|---|---|---|---|---|---|---|---|---|---|
| multi-orderings | 1.00 | 0.96 | 1.00 | 1.00 | n/a | n/a | 1.00 | 0.00 | 1.00 | 0.88 | 1.00 |
| navigate-tree | 0.98 | 0.98 | 0.86 | 0.82 | 0.99 | 0.07 | 0.99 | 0.32 | 1.00 | 0.91 | 1.00 |
| number-checkboxes | n/a | 0.96 | n/a | n/a | 0.84 | 0.26 | 0.99 | 0.00 | n/a | n/a | 0.16 |
| read-table | 1.00 | 0.97 | n/a | n/a | n/a | n/a | 0.97 | 0.01 | n/a | n/a | 0.00 |
| read-table-2 | n/a | 0.95 | n/a | n/a | n/a | n/a | 0.94 | 0.00 | n/a | n/a | 0.00 |
| resize-textarea | n/a | 0.94 | n/a | n/a | 0.99 | 1.00 | 1.00 | 0.27 | n/a | n/a | 0.11 |
| right-angle | n/a | 0.87 | n/a | n/a | 0.97 | 1.00 | 0.98 | 0.26 | n/a | n/a | 0.38 |
| scroll-text | n/a | 0.97 | n/a | n/a | n/a | n/a | 0.96 | 0.04 | n/a | n/a | 0.00 |
| scroll-text-2 | n/a | 0.97 | n/a | n/a | n/a | n/a | 1.00 | 0.88 | n/a | n/a | 0.96 |
| search-engine | 1.00 | 0.97 | 1.00 | 1.00 | n/a | n/a | 1.00 | 0.15 | 0.96 | 0.34 | 1.00 |
| simon-says | n/a | 0.62 | n/a | n/a | n/a | n/a | -0.00 | 0.02 | n/a | n/a | 0.28 |
| simple-algebra | 1.00 | 0.86 | 1.00 | 0.82 | 1.00 | 0.99 | 0.75 | 0.03 | n/a | n/a | 0.04 |
| simple-arithmetic | 1.00 | 0.96 | n/a | n/a | 1.00 | 0.67 | 0.86 | 0.38 | n/a | n/a | 0.07 |
| social-media | 1.00 | 0.96 | 0.98 | 0.82 | n/a | n/a | 0.90 | 0.03 | 1.00 | 0.21 | 1.00 |
| social-media-all | 1.00 | 0.89 | 1.00 | 1.00 | n/a | n/a | 0.75 | 0.00 | 0.31 | 0.00 | 1.00 |
| social-media-some | 1.00 | 0.91 | 0.90 | 0.90 | n/a | n/a | 0.85 | 0.01 | 0.68 | 0.02 | 0.42 |
| terminal | 1.00 | 0.88 | 1.00 | 0.98 | n/a | n/a | -0.01 | 0.00 | n/a | n/a | 0.00 |
| text-editor | n/a | 0.88 | n/a | n/a | n/a | n/a | 0.98 | 0.11 | n/a | n/a | 0.01 |
| text-transform | 0.98 | 0.86 | 0.80 | n/a | 0.92 | 0.91 | 0.60 | 0.19 | n/a | n/a | 0.00 |
| tic-tac-toe | 0.70 | 0.71 | 0.56 | 0.48 | 0.83 | 0.76 | 0.83 | 0.32 | 0.56 | 0.48 | 0.47 |
| unicode-test | 1.00 | 0.99 | n/a | n/a | 1.00 | 0.64 | 1.00 | 0.86 | n/a | n/a | n/a |
| use-autocomplete | 0.98 | 0.98 | 0.58 | 0.88 | 0.99 | 0.95 | 1.00 | 0.07 | 0.98 | 0.22 | 0.98 |
| use-colorwheel | n/a | 0.90 | n/a | n/a | 0.97 | 0.98 | 0.98 | 0.68 | n/a | n/a | 1.00 |
| use-colorwheel-2 | n/a | 0.94 | n/a | n/a | 0.95 | 1.00 | 0.95 | 0.38 | n/a | n/a | 1.00 |
| use-slider | 0.98 | 0.98 | n/a | n/a | 0.92 | 0.69 | 0.91 | 0.18 | n/a | n/a | 0.51 |
| use-slider-2 | n/a | 0.97 | n/a | n/a | 1.00 | 0.09 | 0.95 | 0.03 | n/a | n/a | 0.15 |
| use-spinner | 1.00 | 0.98 | 0.88 | 0.90 | n/a | n/a | 1.00 | 0.47 | 0.11 | 0.07 | 0.17 |
| visual-addition | n/a | 0.97 | n/a | n/a | 1.00 | 0.68 | 0.99 | 0.36 | n/a | n/a | 0.01 |
| **Average** | 0.992 | 0.935 | 0.906 | 0.929 | 0.962 | 0.665 | 0.935 | 0.305 | 0.802 | 0.484 | 0.646 |
| **# of solved tasks** | 64 | 104 | 54 | 53 | 59 | 59 | 104 | 104 | 56 | 56 | 88 |

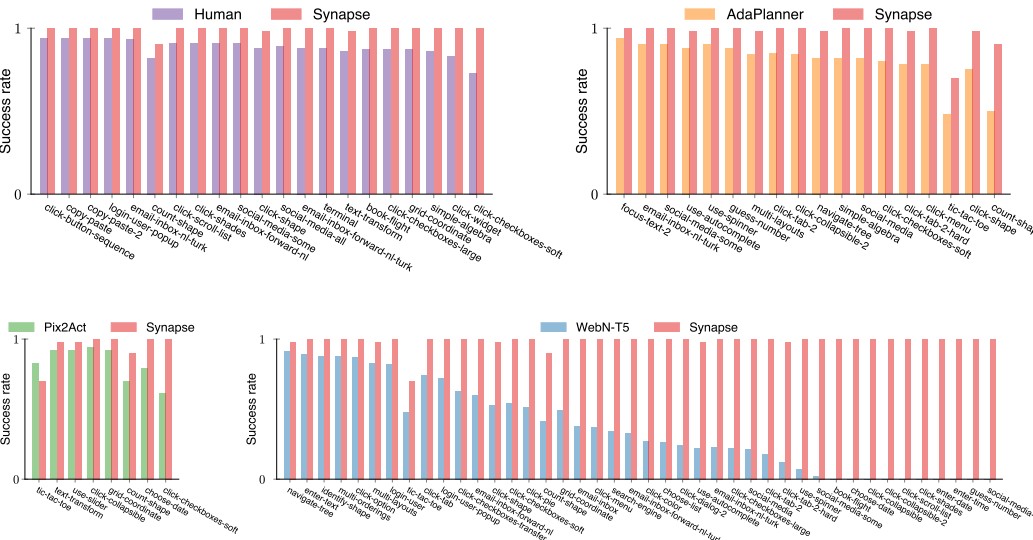

Figure 6: Task-wise comparisons between SYNAPSE, human, two concurrent methods (AdaPlanner and Pix2Act), and WebN-T5, a fine-tuned LLM. Similar to the result in Sec. 4, the bars in each figure are sorted in ascending order based on the success rate differences. Tasks not reported by other methods and those with differences less than 0.05 are not included.

# B ENVIRONMENT DETAILS

## B.1 MINIWOB++

To ensure a fair evaluation, we use the legacy branch of MiniWoB++ as in previous work. Of the 64 tasks that we test (see Tab. 3), 48 are provided with exemplars, and 16 are unseen tasks. The unseen tasks are choose-list, click-checkboxes-transfer, click-checkboxes, click-option, click-tab-2-hard, click-test-2, click-test, copy-paste, login-user, email-inbox-forward-nl-turk, email-inbox-forward-nl, email-inbox, enter-text, multi-orderings, simple-arithmetic, and unicode-test. The average number of exemplars per task for RCI and SYNAPSE is 1.32 and 3.45, respectively. However, the number

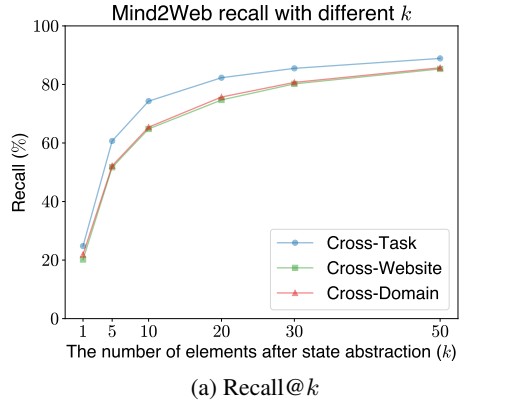
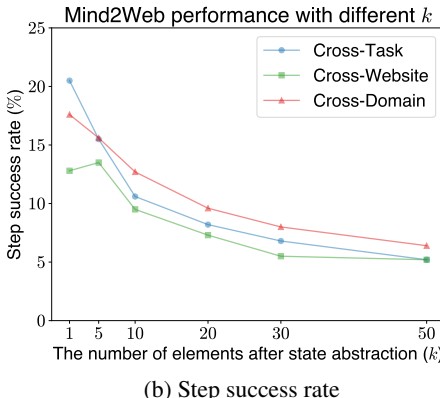

(a) Recall@$k$          (b) Step success rate

Figure 7: Ablations of the number of elements after state abstraction in Mind2Web. The backbone language model here is CodeLlama-Instruct-7B.

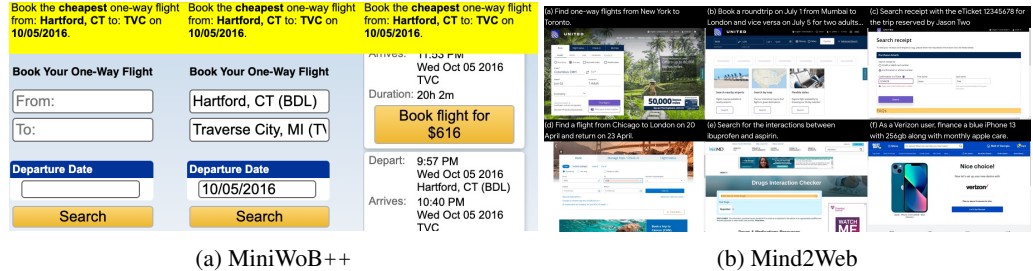

(a) MiniWoB++          (b) Mind2Web

Figure 8: An illustration of MiniWoB++ and Mind2Web.

of exemplars for each task varies. For simple tasks, we use a similar number of exemplars as in RCI. To solve complex tasks (e.g., book-flight), we use more exemplars, which lowers the average data efficiency. For example, in book-flight, we include 5 exemplars for better robustness. Moreover, RCI relies on self-correction (task, state, and agent grounding) while SYNAPSE does not. The success rate of RCI will decrease by around 50% when one of them is missing. Also, one of the benefits of our method is that we can leverage more exemplars in context because of state abstraction.

We use the raw HTML code to represent the agent's state space. We provide the action space of the computer agent in the system prompt. The action space consists of five keyboard and mouse operations, following the prior work (Kim et al., 2023). We use code generation to ground the actions generated by the LLM, the same as the concurrent work (Sun et al., 2023). We use exemplars that specify to click on the first element of all matching ones to avoid clicking on multiple items specified by the xpath. The action space is described in the system prompt:

```
> Role: System
You are a large language model trained to navigate the web. To accomplish the task, use
    methods in the following Agent class to generate actions until you need the new state to
    proceed.
```
class Agent:
    def __init__(self, args):
        ...

    # Action: type a string via the keyboard
    def type(self, characters: str) -> None:
        ...

    # Action: click an HTML element with a valid xpath
    def click_xpath(self, xpath: str):
        ...

    # Actions: press a key on the keyboard, including:
    # enter, space, arrowleft, arrowright, backspace, arrowup, arrowdown, command+a, command+c
    , command+v
```

```
    def press(self, key_type: str) -> None:
        ...

    # Action: click an option HTML element in a list with a valid xpath
    def click_option(self, xpath: str):
        ...

    # Action: move mouse cursor on an HTML element with a valid xpath
    def movemouse(self, xpath: str):
        ...
```

In MiniWoB++, there are two types of state abstraction prompts. For scenarios where the context can handle multiple states, we utilize state-observation pairs. These pairs are collected alongside the trajectories. The states are obtained from the environment, while the cleaned observations are provided by humans. To further reduce human efforts, we can also automatically infer cleaned observations given trajectories using LLMs. In scenarios with complex states where explicit abstraction is not applicable, we implicitly abstract states by pairing task descriptions and state-parsing code. Similarly, the task descriptions are already given by the environment. The code here is collected via zero-shot sampling from LLMs (GPT-4). The code is tested by executing it with states derived from the environment and having its results reviewed by humans. The state abstraction prompts are retrieved from the memory, together with trajectories.

## B.2 MIND2WEB

Mind2Web offers a comprehensive collection of over 2,000 tasks from 137 websites in 31 different domains. The task descriptions in Mind2Web only provide high-level goals. It intentionally avoids detailed, step-by-step instructions, encouraging the agents to autonomously understand and perform tasks, rather than simply following instructions. With Mind2Web, agents can interact with webpages in a way that goes beyond the basic operations of searching or reading. They can click, select, and type into website elements. Such interactions mimic typical human activities on websites, thus expanding the range of tasks the agent can perform.

Mind2Web provides test sets for three levels of generalization: cross-task, cross-websites, and cross-domain. To evaluate cross-task generalization, Mind2Web provides 252 tasks from 69 websites, where the agent has possibly interacted with these websites to perform similar tasks. The cross-website test set consists of 177 tasks from 10 unique websites not seen in the training set but belonging to domains familiar to the agent. This setup tests the agent's ability to navigate new websites within known domains and tasks. The cross-domain test set, containing 912 tasks from 73 websites, requires the agent to adapt to completely unseen domains in the training set. The remaining data, consisting of 1,009 tasks from 73 websites, are used for the training set.

## C PROMPTS

In this section, we provide several examples of our prompting and LLM responses for illustration. We also open source all trajectories in our project website.

### C.1 TERMINAL

To finish the terminal task in MiniWoB++, the agent needs to first list the files and then delete one with a certain extension. By using TaE prompting, SYNAPSE completed this task in four steps: two for state abstraction and two for TaE prompting.

#### C.1.1 STEP 1: STATE ABSTRACTION (EXPLICIT)

**Input:**

```
> Role: User
State:
<div id="wrap" data-wob_ref="2" data-wob_eps="e0">
  <div id="query">Use the terminal below to delete a file ending with the extension .png</div>
  <div id="area" data-wob_ref="3" data-wob_eps="e0"><div data-wob_ref="4" data-wob_eps="e0">
```

```
<div id="terminal" data-wob_ref="5" data-wob_eps="e0">
  <div id="terminal-header" data-wob_ref="6" data-wob_eps="e0">terminal</div>
  <div id="terminal-contents" data-wob_ref="7" data-wob_eps="e0">
    <div class="terminal-output" data-wob_ref="8" data-wob_eps="e0">
  Welcome! Type help for a list of
    available commands.
</div><div class="terminal-output" data-wob_ref="10" data-wob_eps="e0">
  Last login: Tue May 02 2023
</div><div class="terminal-line" data-wob_ref="12" data-wob_eps="e0">
      user$
      
      
    </div>
  </div>
</div>
<input type="text" id="terminal-target" data-wob_ref="15" data-wob_eps="e0">
</div></div>
</div>
```
**Observation:**
Use the terminal below to delete a file ending with the extension .png
**State:**
```
<div id="wrap" data-wob_ref="2" data-wob_eps="e0">
  <div id="query">Use the terminal below to delete a file ending with the extension .py</div>
  <div id="area" data-wob_ref="3" data-wob_eps="e0"><div data-wob_ref="4" data-wob_eps="e0">
<div id="terminal" data-wob_ref="5" data-wob_eps="e0">
  <div id="terminal-header" data-wob_ref="6" data-wob_eps="e0">terminal</div>
  <div id="terminal-contents" data-wob_ref="7" data-wob_eps="e0">
    <div class="terminal-output" data-wob_ref="8" data-wob_eps="e0">
  Welcome! Type help for a list of
    available commands.
</div><div class="terminal-output" data-wob_ref="10" data-wob_eps="e0">
  Last login: Tue May 02 2023
</div><div class="terminal-line" data-wob_ref="12" data-wob_eps="e0">
      user$
      
      
    </div>
  </div>
</div>
<input type="text" id="terminal-target" data-wob_ref="15" data-wob_eps="e0">
</div></div>
</div>
```
**Observation:**
Use the terminal below to delete a file ending with the extension .py
**State:**
```
<div id="wrap" data-wob_ref="2" data-wob_eps="e0">
  <div id="query">Use the terminal below to delete a file that has no file
    extension.</div>
  <div id="area" data-wob_ref="3" data-wob_eps="e0"><div data-wob_ref="4" data-wob_eps="e0">
<div id="terminal" data-wob_ref="5" data-wob_eps="e0">
  <div id="terminal-header" data-wob_ref="6" data-wob_eps="e0">terminal</div>
  <div id="terminal-contents" data-wob_ref="7" data-wob_eps="e0">
    <div class="terminal-output" data-wob_ref="8" data-wob_eps="e0">
  Welcome! Type help for a list of
    available commands.
</div><div class="terminal-output" data-wob_ref="10" data-wob_eps="e0">
  Last login: Tue May 02 2023
</div><div class="terminal-line" data-wob_ref="12" data-wob_eps="e0">
      user$
      
      
    </div>
  </div>
</div>
<input type="text" id="terminal-target" data-wob_ref="15" data-wob_eps="e0">
</div></div>
</div>
```
**Observation:**
Use the terminal below to delete a file that has no file extension.
**State:**
```
<div id="wrap" data-wob_ref="2" data-wob_eps="e0">
  <div id="query">Use the terminal below to delete a file ending with the extension .gpg</div>
  <div id="area" data-wob_ref="3" data-wob_eps="e0"><div data-wob_ref="4" data-wob_eps="e0">
<div id="terminal" data-wob_ref="5" data-wob_eps="e0">
  <div id="terminal-header" data-wob_ref="6" data-wob_eps="e0">terminal</div>
  <div id="terminal-contents" data-wob_ref="7" data-wob_eps="e0">
    <div class="terminal-output" data-wob_ref="8" data-wob_eps="e0">
  Welcome! Type help for a list of
    available commands.
```

```
</div><div class="terminal-output" data-wob_ref="10" data-wob_eps="e0">
  Last login: Sun Sep 24 2023
</div><div class="terminal-line" data-wob_ref="12" data-wob_eps="e0">
      user$
      
      
    </div>
  </div>
</div>
<input type="text" id="terminal-target" data-wob_ref="15" data-wob_eps="e0">
</div></div>
</div>
Observation:
```

## Output:

```
> Role: Assistant
Use the terminal below to delete a file ending with the extension .gpg.
```

### C.1.2    STEP 2: TAE PROMPTING

**Input:**

```
System Prompt (as mentioned in Sec. B.1)
```

Exemplary Trajectory 1

```
> Role: User
Task: Use the terminal below to delete a file ending with the extension .png
Trajectory:
> Role: User
Observation:
Use the terminal below to delete a file ending with the extension .png
Action:
> Role: Assistant
```
agent.type('ls')
agent.press('enter')
```
> Role: User
Observation:
Use the terminal below to delete a file ending with the extension .png
user\$ ls
alloy.png script.zip shark.html sudo.gpg
Action:
> Role: Assistant
```
# Note that it is not allowed to use *.xxx here. alloy.png is the file ending with the
    extension .png. Type the complete filename.
agent.type('rm alloy.png')
agent.press('enter')
```
```

Exemplary Trajectory 2

```
> Role: User
Task: Use the terminal below to delete a file ending with the extension .py
Trajectory:
> Role: User
Observation:
Use the terminal below to delete a file ending with the extension .py
Action:
> Role: Assistant
```
agent.type('ls')
agent.press('enter')
```
> Role: User
Observation:
Use the terminal below to delete a file ending with the extension .py
user\$ ls
bash.gpg image.gif nintendo.py
Action:
> Role: Assistant
```
```

```
# Note that it is not allowed to use *.xxx here.
# nintendo.py is the file ending with the extension .py. Type the complete filename.
agent.type('rm nintendo.py')
agent.press('enter')
```

## Exemplary Trajectory 3

```
> Role: User
Task: Use the terminal below to delete a file that has no file extension.
Trajectory:
> Role: User
Observation:
Use the terminal below to delete a file that has no file extension.
Action:
> Role: Assistant
```
agent.type('ls')
agent.press('enter')
```
> Role: User
Observation:
Use the terminal below to delete a file that has no file extension.
user\$ ls
encrypted.sh chrome.txt search
Action:
> Role: Assistant
```
# Note that it is not allowed to use *.xxx here.
# search is the file that has no file extension while encrypted.sh and chrome.txt are not.
    Type the complete filename.
agent.type('rm search')
agent.press("enter")
```
```

## Current Trajectory

```
> Role: User
Task: Use the terminal below to delete a file ending with the extension .gpg
Trajectory:
> Role: User
Observation:
Use the terminal below to delete a file ending with the extension .gpg
Action:
```

## Output:

```
> Role: Assistant
```
agent.type('ls')
agent.press('enter')
```
```

### C.1.3   STEP 3: STATE ABSTRACTION (EXPLICIT)

### Input:

```
> Role: User
State:
<div id="wrap" data-wob_ref="2" data-wob_eps="e0">
  <div id="query">Use the terminal below to delete a file ending with the extension .png</div>
  <div id="area" data-wob_ref="3" data-wob_eps="e0"><div data-wob_ref="4" data-wob_eps="e0">
<div id="terminal" data-wob_ref="5" data-wob_eps="e0">
  <div id="terminal-header" data-wob_ref="6" data-wob_eps="e0">terminal</div>
  <div id="terminal-contents" data-wob_ref="7" data-wob_eps="e0">
    <div class="terminal-output" data-wob_ref="8" data-wob_eps="e0">
  Welcome! Type help for a list of
    available commands.
</div><div class="terminal-output" data-wob_ref="10" data-wob_eps="e0">
  Last login: Tue May 02 2023
</div><div class="terminal-line" data-wob_ref="17" data-wob_eps="e0">
  user$
  ls
</div><div class="terminal-output" data-wob_ref="20" data-wob_eps="e0">
```

```
      alloy.png script.zip shark.html
        sudo.gpg
</div><div class="terminal-line" data-wob_ref="12" data-wob_eps="e0">
        user$
        
        
      </div>
    </div>
</div>
<input type="text" id="terminal-target" data-wob_ref="15" data-wob_eps="e0">
</div></div>
</div>
Observation:
Use the terminal below to delete a file ending with the extension .png
user\$ ls
alloy.png script.zip shark.html sudo.gpg
State:
<div id="wrap" data-wob_ref="2" data-wob_eps="e0">
  <div id="query">Use the terminal below to delete a file ending with the extension .py</div>
  <div id="area" data-wob_ref="3" data-wob_eps="e0"><div data-wob_ref="4" data-wob_eps="e0">
<div id="terminal" data-wob_ref="5" data-wob_eps="e0">
  <div id="terminal-header" data-wob_ref="6" data-wob_eps="e0">terminal</div>
  <div id="terminal-contents" data-wob_ref="7" data-wob_eps="e0">
    <div class="terminal-output" data-wob_ref="8" data-wob_eps="e0">
  Welcome! Type help for a list of
    available commands.
</div><div class="terminal-output" data-wob_ref="10" data-wob_eps="e0">
  Last login: Tue May 02 2023
</div><div class="terminal-line" data-wob_ref="17" data-wob_eps="e0">
  user$
  ls
</div><div class="terminal-output" data-wob_ref="20" data-wob_eps="e0">
  bash.gpg image.gif nintendo.py</
    span>
</div><div class="terminal-line" data-wob_ref="12" data-wob_eps="e0">
        user$
        
        
      </div>
    </div>
</div>
<input type="text" id="terminal-target" data-wob_ref="15" data-wob_eps="e0">
</div></div>
</div>
Observation:
Use the terminal below to delete a file ending with the extension .py
user\$ ls
bash.gpg image.gif nintendo.py
State:
<div id="wrap" data-wob_ref="2" data-wob_eps="e0">
  <div id="query">Use the terminal below to delete a file that has no file
      extension.</div>
  <div id="area" data-wob_ref="3" data-wob_eps="e0"><div data-wob_ref="4" data-wob_eps="e0">
<div id="terminal" data-wob_ref="5" data-wob_eps="e0">
  <div id="terminal-header" data-wob_ref="6" data-wob_eps="e0">terminal</div>
  <div id="terminal-contents" data-wob_ref="7" data-wob_eps="e0">
    <div class="terminal-output" data-wob_ref="8" data-wob_eps="e0">
  Welcome! Type help for a list of
    available commands.
</div><div class="terminal-output" data-wob_ref="10" data-wob_eps="e0">
  Last login: Tue May 02 2023
</div><div class="terminal-line" data-wob_ref="17" data-wob_eps="e0">
  user$
  ls
</div><div class="terminal-output" data-wob_ref="20" data-wob_eps="e0">
  encrypted.sh chrome.txt search</
    span>
</div><div class="terminal-line" data-wob_ref="12" data-wob_eps="e0">
        user$
        
        
      </div>
    </div>
</div>
<input type="text" id="terminal-target" data-wob_ref="15" data-wob_eps="e0">
</div></div>
</div>
Observation:
Use the terminal below to delete a file that has no file extension.
user\$ ls
```

```
encrypted.sh chrome.txt search
State:
<div id="wrap" data-wob_ref="2" data-wob_eps="e0">
  <div id="query">Use the terminal below to delete a file ending with the extension .gpg</div>
  <div id="area" data-wob_ref="3" data-wob_eps="e0"><div data-wob_ref="4" data-wob_eps="e0">
<div id="terminal" data-wob_ref="5" data-wob_eps="e0">
  <div id="terminal-header" data-wob_ref="6" data-wob_eps="e0">terminal</div>
  <div id="terminal-contents" data-wob_ref="7" data-wob_eps="e0">
    <div class="terminal-output" data-wob_ref="8" data-wob_eps="e0">
  Welcome! Type help for a list of
     available commands.
</div><div class="terminal-output" data-wob_ref="10" data-wob_eps="e0">
  Last login: Sun Sep 24 2023
</div><div class="terminal-line" data-wob_ref="17" data-wob_eps="e0">
  user$
  ls
</div><div class="terminal-output" data-wob_ref="20" data-wob_eps="e0">
  index.rb media.html window.gpg</
     span>
</div><div class="terminal-line" data-wob_ref="12" data-wob_eps="e0">
      user$
      
      
    </div>
  </div>
</div>
<input type="text" id="terminal-target" data-wob_ref="15" data-wob_eps="e0">
</div></div>
</div>
Observation:
```

## Output:

```
> Role: Assistant
Use the terminal below to delete a file ending with the extension .gpg
user\$ ls
index.rb media.html window.gpg
```

### C.1.4   STEP 4: TAE PROMPTING

**Input:**

```
System Prompt (same as step 2)
Exemplary Trajectory 1 (same as step 2)
Exemplary Trajectory 2 (same as step 2)
Exemplary Trajectory 3 (same as step 2)
```

Current Trajectory

```
> Role: User
Task: Use the terminal below to delete a file ending with the extension .gpg
Trajectory:
> Role: User
Observation:
Use the terminal below to delete a file ending with the extension .gpg
Action:
> Role: Assistant
```
agent.type('ls')
agent.press('enter')
```
> Role: User
Observation:
Use the terminal below to delete a file ending with the extension .gpg
user\$ ls
index.rb media.html window.gpg
Action:
```

## Output:

```
> Role: Assistant
```
# Note that it is not allowed to use *.xxx here.
# window.gpg is the file ending with the extension .gpg. Type the complete filename.
```

```
agent.type('rm window.gpg')
agent.press('enter')
```

## C.2 BOOK-FLIGHT

To finish the book-flight task in MiniWoB++, the agent needs to first search the flight and then book one with a certain condition. SYNAPSE completed this task in four steps: two for state abstraction and two for TaE prompting. In step 3, due to the long HTML state, we use implicit state abstraction with code. Temporal abstraction here reduces the number of LLM queries, leading to lower latency and higher success rate in long-horizon tasks.

### C.2.1 STEP 1: STATE ABSTRACTION (EXPLICIT)

**Input:**

```
> Role: User
State:
<div id="wrap" data-wob_ref="2" data-wob_eps="e0">
  <div id="query">Book the cheapest one-way flight from: Hartford, CT to: TVC on 10/05/2016.</div>
  <div id="area" data-wob_ref="3" data-wob_eps="e0">
    <div id="menu" data-wob_ref="4" data-wob_eps="e0">
      <h2 id="header-book" data-wob_ref="5" data-wob_eps="e0">Book Your One-Way Flight</h2>
      <div class="input-container" data-wob_ref="6" data-wob_eps="e0"><input id="flight-from"
    class="flight-input ui-autocomplete-input" type="text" placeholder="From:" autocomplete="
    off" data-wob_ref="7" data-wob_eps="e0"></div>
      <div class="input-container" data-wob_ref="8" data-wob_eps="e0"><input id="flight-to"
    class="flight-input ui-autocomplete-input" type="text" placeholder="To:" autocomplete="
    off" data-wob_ref="9" data-wob_eps="e0"></div>
      <div class="departure-container" data-wob_ref="10" data-wob_eps="e0">
        <div class="departure-header" data-wob_ref="11" data-wob_eps="e0">Departure Date</div>
        <div class="input-container" data-wob_ref="12" data-wob_eps="e0"><input id="datepicker
    " class="flight-input hasDatepicker" type="text" readonly="" data-wob_ref="13" data-
    wob_eps="e0"></div>
      </div>
      <div class="search-container" data-wob_ref="14" data-wob_eps="e0">
        <button id="search" data-wob_ref="15" data-wob_eps="e0">Search</button>
      </div>
    </div>
  <div id="results" class="hide"></div>
  </div>
</div>
Observation:
Type the flight from: 'Hartford, CT' to: '(TVC)' (Hartford, CT is not an airport code. (TVC)
    is an airport code.), and select the date 10/05/2016.
State:
<div id="wrap" data-wob_ref="2" data-wob_eps="e0">
  <div id="query">Book the shortest one-way flight from: MNT to: Islip, NY on 11/05/2016.</div>
  <div id="area" data-wob_ref="3" data-wob_eps="e0">
    <div id="menu" data-wob_ref="4" data-wob_eps="e0">
      <h2 id="header-book" data-wob_ref="5" data-wob_eps="e0">Book Your One-Way Flight</h2>
      <div class="input-container" data-wob_ref="6" data-wob_eps="e0"><input id="flight-from"
    class="flight-input ui-autocomplete-input" type="text" placeholder="From:" autocomplete="
    off" data-wob_ref="7" data-wob_eps="e0"></div>
      <div class="input-container" data-wob_ref="8" data-wob_eps="e0"><input id="flight-to"
    class="flight-input ui-autocomplete-input" type="text" placeholder="To:" autocomplete="
    off" data-wob_ref="9" data-wob_eps="e0"></div>
      <div class="departure-container" data-wob_ref="10" data-wob_eps="e0">
        <div class="departure-header" data-wob_ref="11" data-wob_eps="e0">Departure Date</div>
        <div class="input-container" data-wob_ref="12" data-wob_eps="e0"><input id="datepicker
    " class="flight-input hasDatepicker" type="text" readonly="" data-wob_ref="13" data-
    wob_eps="e0"></div>
      </div>
      <div class="search-container" data-wob_ref="14" data-wob_eps="e0">
        <button id="search" data-wob_ref="15" data-wob_eps="e0">Search</button>
      </div>
    </div>
  <div id="results" class="hide"></div>
  </div>
</div>
Observation:
```

```
Type the flight from: '(MNT)' to: 'Islip, NY' ((MNT) is an airport code. Islip, NY is not an
    airport code.), and select the date 11/05/2016.
State:
<div id="wrap" data-wob_ref="2" data-wob_eps="e0">
  <div id="query">Book the cheapest one-way flight from: Anvik, AK to: MOT on 12/23/2016.</div>
  <div id="area" data-wob_ref="3" data-wob_eps="e0">
    <div id="menu" data-wob_ref="4" data-wob_eps="e0">
      <h2 id="header-book" data-wob_ref="5" data-wob_eps="e0">Book Your One-Way Flight</h2>
      <div class="input-container" data-wob_ref="6" data-wob_eps="e0"><input id="flight-from"
    class="flight-input ui-autocomplete-input" type="text" placeholder="From:" autocomplete="
    off" data-wob_ref="7" data-wob_eps="e0"></div>
      <div class="input-container" data-wob_ref="8" data-wob_eps="e0"><input id="flight-to"
    class="flight-input ui-autocomplete-input" type="text" placeholder="To:" autocomplete="
    off" data-wob_ref="9" data-wob_eps="e0"></div>
      <div class="departure-container" data-wob_ref="10" data-wob_eps="e0">
        <div class="departure-header" data-wob_ref="11" data-wob_eps="e0">Departure Date</div>
        <div class="input-container" data-wob_ref="12" data-wob_eps="e0"><input id="datepicker
    " class="flight-input hasDatepicker" type="text" readonly="" data-wob_ref="13" data-
    wob_eps="e0"></div>
      </div>
      <div class="search-container" data-wob_ref="14" data-wob_eps="e0">
        <button id="search" data-wob_ref="15" data-wob_eps="e0">Search</button>
      </div>
    </div>
  <div id="results" class="hide"></div>
  </div>
</div>
Observation:
```

## Output:

```
> Role: Assistant
Type the flight from: 'Anvik, AK' to: '(MOT)' (MOT is an airport code. Anvik, AK is not an
    airport code.), and select the date 12/23/2016.
```

### C.2.2 STEP 2: TAE PROMPTING

**Input:**

```
System Prompt (same as the terminal task)
```

Exemplary Trajectory 1

```
> Role: User
Task: Book the cheapest one-way flight from: Hartford, CT to: TVC on 10/05/2016.
Trajectory:
> Role: User
Observation:
Type the flight from: 'Hartford, CT' to: '(TVC)' (Hartford, CT is not an airport code. (TVC)
    is an airport code.), and select the date 10/05/2016.
Action:
> Role: Assistant
```
# Type the flight from: 'Hartford, CT' to: '(TVC)'
agent.click_xpath("//*[@id='flight-from']")
agent.type("Hartford, CT")
agent.press("arrowdown")
agent.press("enter")
agent.click_xpath("//*[@id='flight-to']")
agent.type("(TVC)")  # For the airport code, you must use parentheses
agent.press("arrowdown")
agent.press("enter")
# Select the date 10/05/2016
agent.click_xpath("//*[@id='datepicker']")
# Current date: 12/31/2016 (December-October=2). Click on 'Prev' twice
for _ in range(2):
    agent.click_xpath("//*[@title='Prev']")
# Click on the date '5'
agent.click_xpath("//a[text()='5']")
agent.click_xpath("//*[@id='search']")
```
```
> Role: User
Observation:
The price of the cheapest one-way flight is $92
```

**Action:**
**> Role: Assistant**
```
agent.click_xpath("//button[text()='Book flight for $92']")
```

## Exemplary Trajectory 2

**> Role: User**
**Task:** Book the shortest one-way flight from: Abilene, TX to: St Petersburg/Clearwater, FL on
     10/15/2016.
**Trajectory:**
**> Role: User**
**Observation:**
Type the flight from: 'Abilene, TX' to: 'St Petersburg/Clearwater, FL' (Abilene, TX is not an
     airport code. St Petersburg/Clearwater, FL is not an airport code.), and select the date
     10/15/2016.
**Action:**
**> Role: Assistant**
```
# Type the flight from: 'Abilene, TX' to: 'St Petersburg/Clearwater, FL'
agent.click_xpath("//*[@id='flight-from']")
agent.type("Abilene, TX")
agent.press("arrowdown")
agent.press("enter")
agent.click_xpath("//*[@id='flight-to']")
agent.type("St Petersburg/Clearwater, FL")
agent.press("arrowdown")
agent.press("enter")
# Select the date 10/15/2016
agent.click_xpath("//*[@id='datepicker']")
# Current date: 12/31/2016 (December-October=2). Click on 'Prev' twice
for _ in range(2):
    agent.click_xpath("//*[@title='Prev']")
# Click on the date '15'
agent.click_xpath("//a[text()='15']")
agent.click_xpath("//*[@id='search']")
```
**> Role: User**
**Observation:**
The price of the shortest one-way flight is $680
**Action:**
**> Role: Assistant**
```
agent.click_xpath("//button[text()='Book flight for $680']")
```

## Exemplary Trajectory 3

**> Role: User**
**Task:** Book the shortest one-way flight from: MNT to: Islip, NY on 11/05/2016.
**Trajectory:**
**> Role: User**
**Observation:**
Type the flight from: '(MNT)' to: 'Islip, NY' ((MNT) is an airport code. Islip, NY is not an
     airport code.), and select the date 11/05/2016.
**Action:**
**> Role: Assistant**
```
# Type the flight from: '(MNT)' to: 'Islip, NY'
agent.click_xpath("//*[@id='flight-from']")
agent.type("(MNT)")   # For the airport code, you must use parentheses
agent.press("arrowdown")
agent.press("enter")
agent.click_xpath("//*[@id='flight-to']")
agent.type("Islip, NY")
agent.press("arrowdown")
agent.press("enter")
# Select the date 11/05/2016
agent.click_xpath("//*[@id='datepicker']")
# Current date: 12/31/2016 (December-November=1). Click on 'Prev' once
agent.click_xpath("//*[@title='Prev']")
# Click on the date '5'
agent.click_xpath("//a[text()='5']")
agent.click_xpath("//*[@id='search']")
```
**> Role: User**
**Observation:**
The price of the shortest one-way flight is $341

**Action:**
**> Role: Assistant**
```
agent.click_xpath("//button[text()='Book flight for $341']")
```

## Exemplary Trajectory 4

**> Role: User**
**Task:** Book the cheapest one-way flight from: LKE to: Manhattan, KS on 12/12/2016.
**Trajectory:**
**> Role: User**
**Observation:**
Type the flight from: '(LKE)' to: 'Manhattan, KS' ((LKE) is an airport code. Manhattan, KS is
    not an airport code.), and select the date 12/12/2016.
**Action:**
**> Role: Assistant**
```
# Type the flight from: '(LKE)' to: 'Manhattan, KS'
agent.click_xpath("//*[@id='flight-from']")
agent.type("(LKE)")  # For the airport code, you must use parentheses
agent.press("arrowdown")
agent.press("enter")
agent.click_xpath("//*[@id='flight-to']")
agent.type("Manhattan, KS")
agent.press("arrowdown")
agent.press("enter")
# Select the date 12/12/2016
agent.click_xpath("//*[@id='datepicker']")
# Current date: 12/31/2016 (December-December=0). No need to click on 'Prev'
# Click on the date '12'
agent.click_xpath("//a[text()='12']")
agent.click_xpath("//*[@id='search']")
```
**> Role: User**
**Observation:**
The price of the cheapest one-way flight is $152
**Action:**
**> Role: Assistant**
```
agent.click_xpath("//button[text()='Book flight for $152']")
```

## Exemplary Trajectory 5

**> Role: User**
**Task:** Book the cheapest one-way flight from: WMH to: BTR on 11/16/2016.
**Trajectory:**
**> Role: User**
**Observation:**
Type the flight from: '(WMH)' to: '(BTR)' ((WMH) and (BTR) are airport codes.), and select the
    date 11/16/2016.
**Action:**
**> Role: Assistant**
```
# Type the flight from: '(WMH)' to: '(BTR)'
agent.click_xpath("//*[@id='flight-from']")
agent.type("(WMH)")  # For the airport code, you must use parentheses
agent.press("arrowdown")
agent.press("enter")
agent.click_xpath("//*[@id='flight-to']")
agent.type("(BTR)")  # For the airport code, you must use parentheses
agent.press("arrowdown")
agent.press("enter")
# Select the date 11/16/2016
agent.click_xpath("//*[@id='datepicker']")
# Current date: 12/31/2016 (December-November=1). Click on 'Prev' once
agent.click_xpath("//*[@title='Prev']")
# Click on the date '16'
agent.click_xpath("//a[text()='16']")
agent.click_xpath("//*[@id='search']")
```
**> Role: User**
**Observation:**
The price of the cheapest one-way flight is $36
**Action:**
**> Role: Assistant**
```
agent.click_xpath("//button[text()='Book flight for $36']")
```

```
```
```

## Current Trajectory

```
> Role: User
Task: Book the cheapest one-way flight from: Anvik, AK to: MOT on 12/23/2016.
Trajectory:
> Role: User
Observation:
Type the flight from: 'Anvik, AK' to: '(MOT)' (MOT is an airport code. Anvik, AK is not an
    airport code.), and select the date 12/23/2016.
Action:
```

## Output:

```
> Role: Assistant
```
# Type the flight from: 'Anvik, AK' to: '(MOT)'
agent.click_xpath("//*[@id='flight-from']")
agent.type("Anvik, AK")
agent.press("arrowdown")
agent.press("enter")
agent.click_xpath("//*[@id='flight-to']")
agent.type("(MOT)")  # For the airport code, you must use parentheses
agent.press("arrowdown")
agent.press("enter")
# Select the date 12/23/2016
agent.click_xpath("//*[@id='datepicker']")
# Current date: 12/31/2016 (December-December=0). No need to click on 'Prev'
# Click on the date '23'
agent.click_xpath("//a[text()='23']")
agent.click_xpath("//*[@id='search']")
```
```

### C.2.3   STEP 3: STATE ABSTRACTION (IMPLICIT)

**Input:**

```
> Role: User
Filter the following raw HTML state into a clean observation via code based on the task.
State:
<div id="wrap" data-wob_ref="2" data-wob_eps="e0">
  <div id="query">Book the cheapest one-way flight from: Anvik, AK to: MOT on 12/23/2016.</div>
  <div id="area" data-wob_ref="3" data-wob_eps="e0">
    ... (a very long webpage containing flight information)
  </div>
</div>

Write code (between three backticks) to extract the price of the wanted ticket. First check if
    we want shortest or cheapest flight. Parse the HTML using BeautifulSoup. Loop through
    all the available flights. Check time duration (split 'h' and 'm') or flight price based
    on whether we want the shortest or the cheapest. We have searched with airports and the
    date, so there is no need to check them. Keep the integer price of the ticket in a string
     variable 'obs' in this format: 'The price of the {preference} one-way flight is ${price
    }.'. The string of the raw state is already in the variable 'state' so do not repeat the
    state in the code.

Here are some examples:
Write code within three backticks '''' to Book the cheapest one-way flight from: Hartford, CT
    to: TVC on 10/05/2016.
Code:
```python
from bs4 import BeautifulSoup
soup = BeautifulSoup(state, 'html.parser')
preference = 'cheapest'
best_flight_price = float('inf')
# Loop through all the available flights
for flight in soup.find_all("div", class_="flight"):
    # the preference is cheapest, compare flight prices
    price = int(flight.find("button", class_="flight-price")["data-price"])
    if price < best_flight_price:
        best_flight_price = price
price = best_flight_price
obs = f'The price of the {preference} one-way flight is ${price}'
```

```
```
Write code within three backticks '''' to Book the shortest one-way flight from: Abilene, TX
    to: St Petersburg/Clearwater, FL on 10/15/2016.
Code:
```python
from bs4 import BeautifulSoup
soup = BeautifulSoup(state, 'html.parser')
preference = 'shortest'
best_flight_duration = float('inf')
best_flight = None
# Loop through all the available flights
for flight in soup.find_all("div", class_="flight"):
    # the preference is shortest, compare flight durations
    duration = flight.find("div", class_="time-duration").text.split()
    hours = int(duration[0][:-1]) if 'h' in duration[0] else 0
    minutes = int(duration[1][:-1]) if 'm' in duration[1] else 0
    total_minutes = hours * 60 + minutes
    if total_minutes < best_flight_duration:
        best_flight_duration = total_minutes
        best_flight = flight
price = int(best_flight.find("button", class_="flight-price")["data-price"])
obs = f'The price of the {preference} one-way flight is ${price}'
```

Now, write code within three backticks '''' to Book the cheapest one-way flight from: Anvik,
    AK to: MOT on 12/23/2016.
Code:
```

**Output:**

```
> Role: Assistant
```python
from bs4 import BeautifulSoup
soup = BeautifulSoup(state, 'html.parser')
preference = 'cheapest'
best_flight_price = float('inf')
# Loop through all the available flights
for flight in soup.find_all("div", class_="flight"):
    # the preference is cheapest, compare flight prices
    price = int(flight.find("button", class_="flight-price")["data-price"])
    if price < best_flight_price:
        best_flight_price = price
price = best_flight_price
obs = f'The price of the {preference} one-way flight is ${price}.'
```
```

### C.2.4    STEP 4: TAE PROMPTING

**Input:**

```
System Prompt (same as step 2)
Exemplary Trajectory 1 (same as step 2)
Exemplary Trajectory 2 (same as step 2)
Exemplary Trajectory 3 (same as step 2)
Exemplary Trajectory 4 (same as step 2)
Exemplary Trajectory 5 (same as step 2)
```

Current Trajectory

```
> Role: User
Task: Book the cheapest one-way flight from: Anvik, AK to: MOT on 12/23/2016.
Trajectory:
> Role: User
Observation:
Type the flight from: 'Anvik, AK' to: '(MOT)' (MOT is an airport code. Anvik, AK is not an
    airport code.), and select the date 12/23/2016.
Action:
> Role: Assistant
```
# Type the flight from: 'Anvik, AK' to: '(MOT)'
agent.click_xpath("//*[@id='flight-from']")
agent.type("Anvik, AK")
agent.press("arrowdown")
agent.press("enter")
agent.click_xpath("//*[@id='flight-to']")
```

```
agent.type("(MOT)")  # For the airport code, you must use parentheses
agent.press("arrowdown")
agent.press("enter")
# Select the date 12/23/2016
agent.click_xpath("//*[@id='datepicker']")
# Current date: 12/31/2016 (December-December=0). No need to click on 'Prev'
# Click on the date '23'
agent.click_xpath("//a[text()='23']")
agent.click_xpath("//*[@id='search']")
```
**> Role: User**
**Observation:**
The price of the cheapest one-way flight is $65.
**Action:**

**Output:**

```
> Role: Assistant
```
agent.click_xpath("//button[text()='Book flight for $65']")
```
```

### C.3 TEXT-TRANSFORM (FAILURE CASE)

We also demonstrate a failure case in MiniWoB++, where the LLM incorrectly recognizes jrpf as jrfp when typing the transformed text into the text field.

#### C.3.1 STEP 1: TAE PROMPTING

**Input:**

**System Prompt** (same as the terminal task)

Exemplary Trajectory 1

```
> Role: User
Observation:
<div id="wrap" data-wob_ref="2" data-wob_eps="e0">
  <div id="sol" class="rezp" data-wob_ref="3" data-wob_eps="e0"></div>
  <div id="query">Type the text below into the text field and press Submit.</div>
  <div id="area" data-wob_ref="4" data-wob_eps="e0">
    <div id="captcha" data-wob_ref="5" data-wob_eps="e0">rezp</div>
    <div id="form" style="margin-left:13px;margin-top:14px;" data-wob_ref="10" data-wob_eps="
    e0">
      <input type="text" id="tt" data-wob_ref="11" data-wob_eps="e0">
      <button id="subbtn" class="secondary-action" style="margin-top:18px;" data-wob_ref="12"
     data-wob_eps="e0">Submit</button>
    </div>
  </div>
</div>
Action:
> Role: Assistant
```
agent.click_xpath("//*[@id='tt']")  # Locate the input box by clicking on it
agent.type('rezp')  # Type the string in the input box
agent.click_xpath("//*[@id='subbtn']")
```
```

Exemplary Trajectory 2

```
> Role: User
Observation:
<div id="wrap" data-wob_ref="2" data-wob_eps="e0">
  <div id="sol" class="aoy" data-wob_ref="3" data-wob_eps="e0"></div>
  <div id="query">Type the text below into the text field and press Submit.</div>
  <div id="area" data-wob_ref="4" data-wob_eps="e0">
    <div id="captcha" data-wob_ref="5" data-wob_eps="e0">aoy</div>
```

```
    <div id="form" style="margin-left:10px;margin-top:10px;" data-wob_ref="9" data-wob_eps="e0
    ">
      <input type="text" id="tt" data-wob_ref="10" data-wob_eps="e0">
      <button id="subbtn" class="secondary-action" style="margin-top:17px;" data-wob_ref="11"
    data-wob_eps="e0">Submit</button>
    </div>
  </div>
</div>
Action:
> Role: Assistant
```
agent.click_xpath("//*[@id='tt']")   # Locate the input box by clicking on it
agent.type('aoy')   # Type the string in the input box
agent.click_xpath("//*[@id='subbtn']")
```
```

Current Trajectory

```
> Role: User
Observation:
<div id="wrap" data-wob_ref="2" data-wob_eps="e0">
  <div id="sol" class="jrpf" data-wob_ref="3" data-wob_eps="e0"></div>
  <div id="query">Type the text below into the text field and press Submit.</div>
  <div id="area" data-wob_ref="4" data-wob_eps="e0">
    <div id="captcha" data-wob_ref="5" data-wob_eps="e0">jrpf</div>
    <div id="form" style="margin-left:13px;margin-top:19px;" data-wob_ref="10" data-wob_eps="
    e0">
      <input type="text" id="tt" data-wob_ref="11" data-wob_eps="e0">
      <button id="subbtn" class="secondary-action" style="margin-top:16px;" data-wob_ref="12"
    data-wob_eps="e0">Submit</button>
    </div>
  </div>
</div>
Action:
```

**Output:**

```
> Role: Assistant
```
agent.click_xpath("//*[@id='tt']")   # Locate the input box by clicking on it
agent.type('jrfp')   # Type the string in the input box
agent.click_xpath("//*[@id='subbtn']")
```
```

## C.4   MIND2WEB

We only demonstrate one step in Mind2Web because real-world tasks and websites are complicated. The observations are obtained using their element-ranking model. We set $k$ to 3 for the trajectories in memory and previous observations in current history and $k = 5$ for the current observation.

**Input:**

```
> Role: System
You are a large language model trained to navigate the web. Output the next action and wait
    for the next observation. Here is the action space:
1. `CLICK [id]`: Click on an HTML element with its id.
2. `TYPE [id] [value]`: Type a string into the element with the id.
3. `SELECT [id] [value]`: Select a value for an HTML element by its id.
```

Exemplary Trajectory 1

```
> Role: User
Task: Remove the SSD on my cart
Trajectory:
Observation: `<html> <div> <a id=131 shopping cart> <i icon of shopping cart /> <div>  3
     Items </div> <div> $52.97 </div> </a> <ul> <a id=10012 computer peripherals> <
    div> Computer Peripherals </div> </a> <a id=10523 electronics> <div> Electronics </div>
    </a> </ul> </div> </html>`
> Role: Assistant
Action: `CLICK [131]` ([link]  Shopping Cart -> CLICK)
```

> **Role: User**
**Observation:** `<html> <div> <div>  <button id=18445 button> <i trash /> Remove </button> </div> <div> <a view details> SAMSUNG 980 PRO M.2 2280 2TB PCIe Gen 4.0 x4, NVMe 1.3c Samsung V-NAND Internal Solid State Drive (SSD) MZ-V8P2T0B/AM </a> <a id=27481> Internal SSDs </a> </div> </div> </html>`
> **Role: Assistant**
**Action:** `CLICK [18445]` ([button] trash REMOVE -> CLICK)

## Exemplary Trajectory 2

> **Role: User**
**Task:** Add Western Digital internal SSD with 1TB storage and 8GB Ram DDR to the cart.
**Trajectory:**
**Observation:** `<html> <div> <input id=145 search search site /> <ul> <a id=10059 computer peripherals> <div> Computer Peripherals </div> </a> <a id=10570 electronics> <div> Electronics </div> </a> </ul> </div> </html>`
> **Role: Assistant**
**Action:** `CLICK [145]` ([searchbox] Search Site -> CLICK)
> **Role: User**
**Observation:** `<html> <div> <input id=36915 search search site /> <ul> <a id=46898 computer peripherals> <div> Computer Peripherals </div> </a> <a id=47409 electronics> <div> Electronics </div> </a> </ul> </div> </html>`
> **Role: Assistant**
**Action:** `TYPE [36915] [Western Digital internal SSD 1 TB]` ([searchbox] Search Site -> TYPE: Western Digital internal SSD 1 TB)
> **Role: User**
**Observation:** `<html> <div> <div> <input id=81881 search search site western digital internal ssd 1 /> <button id=73846>  Search  </button> </div> <a western digital>  </a> </div> </html>`
...

## Exemplary Trajectory 3

> **Role: User**
**Task:** Upgrade the count of the current SSD in my cart to 10
**Trajectory:**
**Observation:** `<html> <div> <a id=130 shopping cart> <i icon of shopping cart /> <div>  3  Items </div> <div> $85.97 </div> </a> <ul> <a id=10016 computer peripherals> <div> Computer Peripherals </div> </a> <a id=10527 electronics> <div> Electronics </div> </a> </ul> </div> </html>`
> **Role: Assistant**
**Action:** `CLICK [130]` ([link] Shopping Cart -> CLICK)
> **Role: User**
**Observation:** `<html> <div> <div> <a id=26807> Add </a> <a> Edit </a> <a> Done </a> </div> <div> <div> <input id=18580 text qty 1 /> <div> Limit 20 </div> </div> <div> <a id=26918> Add </a> <a> Edit </a> <a> Done </a> </div> </div> </div> </html>`
> **Role: Assistant**
**Action:** `TYPE [18580] [10]` ([textbox] qty -> TYPE: 10)
> **Role: User**
**Observation:** `<html> <div> <div> <a id=46341> Add </a> <a> Edit </a> <a> Done </a> </div> <div> <div> <input id=38114 text qty 10 /> <div> Limit 20 </div> </div> <div> <a id=46473> Add </a> <a> Edit </a> <a> Done </a> </div> </div> </div> </html>`
> **Role: Assistant**
**Action:** `CLICK [38114]` ([textbox] qty -> ENTER)

### Current Trajectory

> **Role: User**
**Task:** Add the cheapest SSD to my cart
**Trajectory:**
**Observation:** `<html> <div> <div> <i id=104 site menu /> <input id=99 search search site /> </div> <ul> <a id=9139 computer systems> <div> Computer Systems </div> </a> <a id=10010 computer peripherals> <div> Computer Peripherals </div> </a> <a id=10521 electronics> <div> Electronics </div> </a> </ul> </div> </html>`

## Output:

> **Role: Assistant**
**Action:** `CLICK [9139]` ([link] Computer Systems -> CLICK)

