# OpenReview forum: "Synapse: Trajectory-as-Exemplar Prompting with Memory for Computer Control"
_ICLR.cc/2024/Conference — ICLR 2024 poster_

### Official Review · Reviewer_bu8M · 2023-10-31

**Soundness:** 3 good
**Presentation:** 3 good
**Contribution:** 2 fair
**Rating:** 8
**Confidence:** 4

**Summary:**

This paper studies the problem of data driven computer control using large language models (LLM's). It adds three components to improve capabilities of current approaches. It abstracts from the state of the environment by parsing it via an LLM such that only information necessary for the task is kept. It conditions on complete trajectories of similar tasks. It keeps a memory from which similar example trajectories are sampled.  The method is compared against strong baselines on two current benchmarks in the field of data-driven computer control. It shows improved task success rate on both, while often needing less data. Ablations on each component of the method show that the state abstraction procedure is the main reason for the improved performance.

**Strengths:**

The method is mostly explained well and it is easy to follow the explanation. Experiments and results are presented well.

Improved task success rate on two benchmarks in the field of data driven computer control.

Competitive baselines from a variety of recent approaches in the domain of data driven computer control.

Ablations investigate the effect of each component of the method.

The state abstraction approach allows the method to be tested on tasks that were infeasible for prior methods.

**Weaknesses:**

- From the results in the MindAct task it seems that adding any trajectories helps but it does not matter which ones.

- Have you studied the effect on performance on the MiniWob++ environment for different amount of tasks in the memory. The comparison to RCI does not seem fair at the moment (Figure 3)?

- It would be helpful to have an illustrating example of the result of the state abstraction procedure (i.e. a figure of a (state,observation)-pair) as well as an example prompt (maybe in the Appendix). Figure 1 only shows that some <body> and <h1> tag got removed but it's not yet clear to me what the result of the state abstraction procedure looks like. Its also not clear to me where the examples from the state abstraction prompt come from. Are they designed a-priori? If yes how and how difficult is that? Are the same examples reused for all states or also taken from the memory?

- The text states (last paragraph of 4.3) that failed trajectories are displayed in Appendix C, but Appendix C only refers to the supplementary material?

- In the original paper of MindAct, they used 3 in-context exemplars for GPT3.5  but Synapse uses 5, so the comparison favours Synapse. In general I would maybe add a sentence that clarifies that results are SOTA, conditioned on the fact that GPT3.5 was used as the underlying LLM.

**Questions:**

- Have you considered building up the memory from scratch by adding successful trajectories over the course of training?

---

> ### Author Response · Authors · 2023-11-17
> **Response to Reviewer bu8M (1/2)**
>
> We sincerely thank the reviewer for reading our paper and providing valuable insights.
>
> > **Q1.** From the results in the MindAct task it seems that adding any trajectories helps but it does not matter which ones.
>
> **A1.** It seems that there are some misunderstandings in our Mind2Web results. Let us elaborate on the results here.
>
> - In `SYNAPSE w/ state abstraction`, we simply adopt state abstraction principles to direct generation (with fewer top-ranked elements in clean observations), which outperforms the MCQ-formatted prompting used in MindAct. Here, we use the same exemplars as those used in the MindAct, but the format is changed from selecting an answer from multiple choices to directly generating the action, and we do not recursively select one answer from five candidates until only one is left in the top-50 elements. Instead, we directly generate the action based on abstracted states with the top-5 elements.
> - The `SYNAPSE w/ state abstraction + TaE` variant integrates TaE prompting on top of state abstraction, where we replace the original prompting format with trajectories. The results show the benefits of using trajectories as exemplars. In both these two variants, we use **static** few-shot exemplars, i.e., the exemplars are the same for all tasks during evaluation. This is the standard paradigm of few-shot learning, where we provide examples of input and output for LLMs. The performance lift does not result from "adding any trajectories". It is due to the change in the exemplar format.
> - In `SYNAPSE w/ state abstraction + TaE + memory`, we encode the training set as the memory and apply similarity search to match trajectories from similar tasks, which continues to improve performance. This also showcases that adding relevant trajectories as exemplars can be beneficial compared to "adding any trajectories".
>
> We have updated the caption of Table 1 to clarify these details.
>
> > **Q2.** Have you studied the effect on performance on the MiniWob++ environment for different amount of tasks in the memory. The comparison to RCI does not seem fair at the moment (Figure 3)?
>
> **A2.** It is noteworthy that RCI solves 54 tasks with exemplars from 54 tasks, while Synapse solves 64 tasks with exemplars from 48 tasks, including challenging ones that RCI cannot solve, such as book-flight. The average number of exemplars per task for RCI and Synapse is 1.32 and 3.45, respectively. However, the number of exemplars for each task varies. For simple tasks, we use a similar number of exemplars as in RCI. To solve complex tasks (e.g., book-flight), we use more exemplars, which lowers the average data efficiency. For example, in book-flight, we include 5 exemplars for better robustness. Moreover, RCI relies on self-correction and our method does not. They proposed several self-correction modules (task/state/agent grounding), and their success rate will decrease by around 50% without either one of them, according to their paper [1]. Also, one of the benefits of our method is that we can leverage more exemplars in-context because of state abstraction.

---

> ### Author Response · Authors · 2023-11-17
> **Response to Reviewer bu8M (2/2)**
>
> > **Q3.** It would be helpful to have an illustrating example of the result of the state abstraction procedure (i.e. a figure of a (state,observation)-pair) as well as an example prompt (maybe in the Appendix). Figure 1 only shows that some `<body>` and `<h1>` tag got removed but it's not yet clear to me what the result of the state abstraction procedure looks like. Its also not clear to me where the examples from the state abstraction prompt come from. Are they designed a-priori? If yes how and how difficult is that? Are the same examples reused for all states or also taken from the memory?
>
> **A3.** We have reorganized Sec. 3.2 to clarify the state abstraction procedure and included more examples in Appendix C for illustrations. Fig. 1 aims to show that raw states can be very long-winded, which we omitted in  `<h1>…` , while clean observations are much more concise with only task-relevant information, like `<input id="where" type="text">`.
>
> In MiniWoB++, there are two types of state abstraction prompts:
>
> - For scenarios where the context can handle multiple states, we utilize state-observation pairs ⟨state, observation⟩. These pairs are collected alongside the trajectories. The states are obtained from the environment, while the cleaned observations are provided by humans. To further reduce human efforts, we can also automatically infer cleaned observations given trajectories using LLMs [2].
>
> - In scenarios with complex states where explicit abstraction is not applicable, we implicitly abstract states by pairing task descriptions and state-parsing code, denoted as ⟨task, code⟩. Similarly, the task descriptions are already given by the environment. The code here is collected via zero-shot sampling from LLMs (GPT-4) plus validation. This data collection pipeline can also be augmented with LLMs to reduce human effort.
>
> The state abstraction prompts are retrieved from the memory, together with action generation prompts (trajectories).
>
> In fact, one of the advantages of trajectory-as-exemplars over other methods is that it is simple and scalable to convert human demonstrations into exemplars (as they are already trajectories), without much human effort in designing promptings like multi-choice question-answering or high-level semantic plans in natural language.
>
> > **Q4.** The text states (last paragraph of 4.3) that failed trajectories are displayed in Appendix C, but Appendix C only refers to the supplementary material?
>
> **A4.** We have added trajectories of both successful (terminal and book-flight) and failed (text-transform and Mind2Web) cases in Appendix C.
>
> > **Q5.** In the original paper of MindAct, they used 3 in-context exemplars for GPT3.5 but Synapse uses 5, so the comparison favours Synapse. In general I would maybe add a sentence that clarifies that results are SOTA, conditioned on the fact that GPT3.5 was used as the underlying LLM.
>
> **A5.** The comparisons are fair because Synapse also uses three in-context exemplars, the same number of exemplars as MindAct. The $k=5$ refers to the number of elements after state abstraction. The reviewer's misunderstanding may come from our abuse of the notation $k$ for both top-$k$ memory retrieval and top-$k$ state abstraction. We have updated the notation for memory to top-$n$ to avoid misunderstanding.
>
> To investigate how Synapse performs on other language models, we added the Mind2Web experiments on CodeLlama-7B in Table 1, where Synapse outperforms MindAct with a 2.5x average step success rate across three levels of generalization (cross-task, cross-website, and cross-domain). These results are consistent with the results with GPT-3.5 as the underlying LLM.
>
> > **Q6.** Have you considered building up the memory from scratch by adding successful trajectories over the course of training?
>
> **A6.** Thank you for this suggestion. It would be interesting to explore this aspect. However, in this work, we mainly investigate the design choices around few-shot exemplars, including state abstraction, trajectory-as-exemplar, and exemplar memory. Therefore, we initialize the memory with existing exemplars and fix it during experiments. In fact, Synapse complements existing methods that build up memory during the agent's interaction with the environment [3]. Specifically, our method can guide the collection of exemplars and the format to store them. It is definitely interesting to integrate trajectory exemplars into open-ended learning agents. Another direction for future work is to query human intervention and add their demonstrations into the memory when the similarity score is low, i.e., the agent is faced with unfamiliar situations.
>
> [1] Kim et al. "Language models can solve computer tasks." NeurIPS 2023.
>
> [2] He et al. "Annollm: Making large language models to be better crowdsourced annotators." *arXiv preprint arXiv:2303.16854* (2023).
>
> [3] Wang et al. "Voyager: An open-ended embodied agent with large language models." arXiv preprint arXiv:2305.16291 (2023).

---

> ### Comment · Reviewer_bu8M · 2023-11-20
>
> Thank you for your detailed replies.
>
> A1: Maybe "adding" was not a very precise formulation but I think my main point is that the relative performance improvement from switching to trajectories as exemplars is large while the performance improvement to choosing trajectories dynamically via the memory is marginal.
>
> A2: I think this makes it clear to me that the comparison is fair. I would maybe add a sentence that the difference in exemplars per task between RCI and Synapse comes from using more exemplars for difficult tasks that RCI cannot solve.
>
> A3: It could be valuable to have some overview of human effort required by different approaches (maybe in the Appendix) to  make it clear on how much engineering each of the approaches relies on. The validation part when GPT-4 is required to produce code is done by humans or by running the code on some tests?
>
> A4: I think, that makes it very clear how large the reduction in the state representation is due to abstraction.
>
> A5: Yes, I mixed up these two values, thanks.
>
> A6: Thanks.
>
> Since my main concerns have been resolved, I will update my rating.

---

> > ### Author Response · Authors · 2023-11-23
> > **Thank you for your update!**
> >
> > Thank you for your feedback and for updating the rating! We are glad to hear that your main concerns have been resolved.
> >
> > **A1.** The benefits of exemplar memory are two-fold: 1) matching exemplars for new tasks, and 2) improving performance using exemplars from similar tasks.
> >
> > In MiniWoB++, memory is particularly helpful because many tasks share similar web layouts. It allows us to solve 64 tasks using exemplars from 48 tasks, while previous methods require specific exemplars for each task because they do not have the mechanisms to match exemplars with given tasks.
> >
> > In Mind2Web, memory's contribution is relatively marginal, because the tasks, websites, and even domains in the test sets are completely different from the training set (memory). However, in real-world use cases, when the similarity distance is large, we can query human demonstration and add it to the memory (as discussed in **A6**). We demonstrate this with a proof-of-concept experiment on the first 20 tasks of Mind2Web training set. Here, the retrieved exemplars are more similar to the tasks, leading to better performance improvements.
> >
> > | Method                                      | Ele. Acc (%) | Step SR (%) |
> > | ------------------------------------------- | ------------ | ----------- |
> > | GPT-3.5 w/ state abstraction + TaE          | 32.6         | 29.4        |
> > | GPT-3.5 w/ state abstraction + TaE + memory | **44.5**     | **39.9**    |
> >
> > **A2 & A3.** We have updated these details in Appendix B.
> >
> > We hope the above discussion addresses your concerns. Please let us know if you have any other questions.

---

### Official Review · Reviewer_mGYB · 2023-11-07

**Soundness:** 3 good
**Presentation:** 3 good
**Contribution:** 3 good
**Rating:** 6
**Confidence:** 2

**Summary:**

This paper proposes a new technique to prompt language models with example trajectories for computer control tasks. It utilizes language models to abstract clean states for long-context conditioning, and design a retrieval module to improve generalization.

**Strengths:**

1. The writing and presentation is clear in general
2. The proposed method surpassed baselines like CC-Net, RCI on various benchmark like MiniWoB++ and Mind2Web

**Weaknesses:**

1. Some details are not well explained, see questions.

**Questions:**

1. SYNAPSE uses state abstraction to make sure LLM can learn from compact information while ensuring the context length is well utilized. However, in the experiments, SYNAPSE sets k=5 for retrieval. How does the performance change if k grows? And why longer context is not helping here?

2. Table 1 seems confusing. It's non-standard to use "SYNAPSE w/ state abstraction, SYNAPSE w/ trajectory-as-exemplar, SYNAPSE w/ training set as memory" to indicate gradually adding "state abstraction, trajectory-as-exemplar, training set as memory" 3 components. If you want to demonstrate a component *A* is useful, better to write it in "SYNAPSE w/o *A*", as SYNAPSE is a method that combines all components

3. For Table 1 training set as memory experiments, do you add all the training data for 3 scenarios ("cross-task, cross-web, and cross-domain)? Or only the corresponding training data (e.g. cross-task but same web when evaluated on cross-task setting).

4. For tasks that need code to abstract state information, how can you examine if a code can successfully extract desired information? if examined from task success, it can be because of the suboptimal execution of policy.

5. For the cross-domain tasks, retrieval doesn't seem to help, can you give a better explanation about this result? The retrieval number is set to 5, which is a small number. Is it because the retriever doesn't return the desired examples? Maybe showing some retrieved examples can be more intuitive.

6. "The action generation starts from prompting LLMs with successful trajectories to warm up LLMs
with the dynamics of the current environment". Is it describing prompting with retrieved trajectories? Or that's some other warmup procedure? If it's cross-task/cross-web/cross-domain setting, how's current environment defined?

---

> ### Author Response · Authors · 2023-11-17
> **Response to Reviewer mGYB (1/2)**
>
> We sincerely thank the reviewer for reading our paper and providing valuable insights.
>
> > **Q1.** SYNAPSE uses state abstraction to make sure LLM can learn from compact information while ensuring the context length is well utilized. However, in the experiments, SYNAPSE sets k=5 for retrieval. How does the performance change if k grows? And why longer context is not helping here?
>
> **A1.** As suggested by the reviewer, we have added Fig. 7 in the appendix to ablate the performance with different $k$ (the number of elements in the observation after abstraction). The performance drops as $k$ grows, even though the recall@$k$ becomes higher (the probability of the correct element appearing in the observation). There are two possible explanations: i) more irrelevant information poses challenges in selecting correct ones [1,2], ii) current LLMs struggle to handle long context, e.g., the performance gets worse as the input context grows longer [3].
>
> > **Q2.** Table 1 seems confusing. It's non-standard to use "SYNAPSE w/ state abstraction, SYNAPSE w/ trajectory-as-exemplar, SYNAPSE w/ training set as memory" to indicate gradually adding "state abstraction, trajectory-as-exemplar, training set as memory" 3 components. If you want to demonstrate a component A is useful, better to write it in "SYNAPSE w/o A", as SYNAPSE is a method that combines all components
>
> **A2.** Synapse integrates the three components—state abstraction, trajectory-as-exemplar, and memory in a specific sequence. State abstraction shrinks the length of states and thereby trajectories, which is the prerequisite for trajectory-as-exemplar prompting. Similarly, the memory stores exemplary trajectories as basic units. We believe that the current results of gradually adding components are enough to validate the effectiveness of the three components.
>
> > **Q3.** For Table 1 training set as memory experiments, do you add all the training data for 3 scenarios ("cross-task, cross-web, and cross-domain)? Or only the corresponding training data (e.g. cross-task but same web when evaluated on cross-task setting).
>
> **A3.** The Mind2Web dataset is split into a training set and three test sets: Cross-Task, Cross-Website, and Cross-Domain, to evaluate generalizability over tasks from the same websites, unseen websites from similar domains, and completely unseen domains in the training set, respectively. We only store the training set in the memory. All three cross-* test sets are not stored in the memory. We have added these details in Sec. 4.1, and provided detailed descriptions of these dataset splits in Appendix B.2.
>
> > **Q4.** For tasks that need code to abstract state information, how can you examine if a code can successfully extract desired information? if examined from task success, it can be because of the suboptimal execution of policy.
>
> **A4.** If the code execution raises an error, we ask the LLM to perform zero-shot
> state abstraction, i.e., directly abstracting state information without any exemplars. For simplicity, we did not introduce any success detection or self-correction to examine what components caused the failure, which is not our core contribution. When the task fails, we directly stop and mark the task as a failure. This detail has been updated in Sec. 3.2.

---

> ### Author Response · Authors · 2023-11-17
> **Response to Reviewer mGYB (2/2)**
>
> > **Q5.** For the cross-domain tasks, retrieval doesn't seem to help, can you give a better explanation about this result? The retrieval number is set to 5, which is a small number. Is it because the retriever doesn't return the desired examples? Maybe showing some retrieved examples can be more intuitive.
>
> **A5.** After carefully examining the results, we found that there is a minor calculation error. The results have been updated in Table 1. However, the conclusion that memory does not help much in cross-domain tasks still holds. We also added experiments on CodeLlama-7B to validate the conclusion. We have added Table 2 in Sec. 4.5 to visualize the distance between retrieved exemplars and the tasks for cross-task, cross-website, and cross-domain test sets. As shown in Table 2, as the distance becomes larger, i.e., less similarity, the performance gain ($\Delta$Step SR) becomes worse. As discussed in Sec. 4.5, this might be attributed to the LLM being influenced by exemplars from completely unrelated domains.
>
> The number of few-shot exemplars is actually 3, the same as used in the baseline. We noticed that we abused the notation $k$ for both the top-$k$ parameter in memory retrieval and the top-$k$ parameter in Mind2Web state abstraction, and we have updated the notation for memory retrieval to top-$n$ to avoid misunderstanding. Intuitively, deciding the number of exemplars is a tradeoff between performance and context limits. Adding more exemplars can lead to better performance, but it also uses more tokens. As discussed in A1, current LLMs have limited context and struggle to perform well when the context is long. Three-shot learning is a suitable balance, ensuring a fair comparison with the baseline and taking into account both sides of the issue.
>
> > **Q6.** "The action generation starts from prompting LLMs with successful trajectories to warm up LLMs with the dynamics of the current environment". Is it describing prompting with retrieved trajectories? Or that's some other warmup procedure? If it's cross-task/cross-web/cross-domain setting, how's current environment defined?
>
> **A6.** We only prompt with retrieved exemplary trajectories. There is no other warmup procedure. The environment here refers to the observation and action space. In cross-task/cross-website/cross-domain settings, these spaces remain consistent. To avoid misunderstanding, we have deleted this expression from the paper.
>
> [1] Gur et al. "A real-world webagent with planning, long context understanding, and program synthesis." arXiv preprint arXiv:2307.12856 (2023).
>
> [2] Deng et al. "Mind2Web: Towards a Generalist Agent for the Web." NeurIPS 2023.
>
> [3] Liu et al. "Lost in the middle: How language models use long contexts." arXiv preprint arXiv:2307.03172 (2023).

---

### Official Review · Reviewer_z7sh · 2023-11-08

**Soundness:** 3 good
**Presentation:** 3 good
**Contribution:** 3 good
**Rating:** 8
**Confidence:** 3

**Summary:**

This paper presents a prompting-based method (SYNAPSE) for computer control that takes in states and outputs actions in a discrete, human-computer-interaction-like action space. The paper is motivated by gaps in existing computer control methods: in-context learning, which struggles with long-horizon tasks and generalization and often needs post-hoc correction; and trained/fine-tuned methods, which are data- and compute-inefficient.

SYNAPSE consists of three components: a state abstraction procedure in which an LLM is used to extract important information from the raw state, thereby significantly reducing token length; "trajectories as exemplars" (TaE) prompting, in which full, relevant trajectories (sequences of clean states and actions) are added to the prompt for generating each new action; and exemplar memory in which trajectories and their associated tasks are encoded and stored, to be used as exemplar trajectories when a related task is being planned.

Experiments are conducted on MiniWoB++ and Mind2Web. The MiniWoB++ experiments show SYNAPSE's major data efficiency (and moderate performance) gains over BC+RL and finetuning baselines, as well as performance and simplicity gains over ICL baselines. The Mind2Web experiments demonstrate the same, along with its generalization capability across tasks, websites, and domains. SYNAPSE beats SOTA on various tasks from MiniWoB++, including notably difficult ones for which human-level performance is not achieved by prior work.

**Strengths:**

#### Quality
- Method is intuitive and effective
- Experimental setup is useful. It is useful to have one standard benchmark with extensive comparison to baselines and one realistic benchmark on which more qualitative advantages like generalization are shown.
- Results are promising - the similarity to human performance on MiniWoB++ and the significant increases on generalization show the power of SYNAPSE's TaE and exemplar memory components.
- Ablations are useful as well to concretely break down each component of the approach.
- Overall a well-constructed, well-executed paper.

#### Clarity
- Very well written!
- Figures are especially useful compared to a lot of papers. Tables as well. Overall, results are not only promising but well-communicated.

#### Originality
The paper positions itself well in prior work. As far as I am aware, the specific three components used here have not been combined elsewhere.

#### Significance
The performance on existing computer control benchmarks is impressive, especially when comparing to human baselines. Figure 4 particularly tells a story of a significant improvement in capability. The results on MiniWoB++ show only some improvement, so it's unclear how significantly SYNAPSE distinguishes itself, but the results on Mind2Web show a lot of improvement margin.

**Weaknesses:**

#### Quality
- The same ideas (gaps in prior work, the three components and their descriptions, the benchmarks being used) are repeated several times throughout the paper. While I very much agree that some amount of repetition is crucial to get a reader to understand what you are saying, in this case it's not just two or three times but four or five, or even if it's three, there's lots of detail every time. It feels as though the repetition is there to fill up space, even if that's not really the case. It gets tiring to read the same high-level concepts over and over.

#### Clarity
- I don't quite understand what "step success rate" is as opposed to "success rate" - it's brought up several times as the metric for Mind2Act, but not defined as far as I'm aware.
- It doesn't seem to be made clear what exactly the state abstraction entails. In figure 1, the only difference between "raw state" and "clean state" is `<body><h1>…</h1></body>” --> <input id=”where” type=”text”>`. There aren't clear examples in the main text, but intuition about this would be very useful.
- Not clear how SYNAPSE's history-in-prompt approach is significantly different from Mind2Act's (or lots of other works') to the point of being novel.
- From both Fig 1 and the text, it's unclear when new trajectory exemplars and example state abstractions are generated - is it per task (that's what I'd think) or per step?
- Fig 3: might help to have clearer visual distinction of SYNAPSE and the human baseline, especially since the colors aren't always visible. Lines/shading/callout boxes could help.

#### Originality
Though the approach is technically unique and clearly effective, there are some originality concerns in terms of what the paper claims. For example, the paper positions itself against works like Inner Monologues and SayCan by saying that they focus on high-level planning; however, though RCI uses *even higher* level planning, SYNAPSE is still working with a discrete, abstract action space. Meanwhile, the low-level BC/RL/motion planning+control used for the robotics papers cited are operating in a much more granular, potentially continuous, action space. For another example, the paper spends considerable space discussing how it is unique compared to other prompting-based methods in helping the LLM understand the current state. However, MindAct gives trajectory-so-far back to the LLM when prompting for the next step. SYNAPSE gives observation as well, thereby informing the LLM in more detail about the current state, this just isn't a very original idea. It's an important step past MindAct, but not a big one, especially relative to how much it's discussed as a unique contribution.


Nits:
- Page 5, last paragraph: `Obseravtion` is a misspelling

**Questions:**

- What exactly is step success rate vs. success rate?
- It is inherently interesting that this approach does not need self-correction unlike other prompting-based approaches, but does that offer any significant performance/efficiency gain?

---

> ### Author Response · Authors · 2023-11-17
> **Response to Reviewer z7sh (1/3)**
>
> We sincerely thank the reviewer for reading our paper and providing valuable insights.
>
> > **Q1.** The same ideas (gaps in prior work, the three components and their descriptions, the benchmarks being used) are repeated several times throughout the paper. While I very much agree that some amount of repetition is crucial to get a reader to understand what you are saying, in this case it's not just two or three times but four or five, or even if it's three, there's lots of detail every time. It feels as though the repetition is there to fill up space, even if that's not really the case. It gets tiring to read the same high-level concepts over and over.
>
> **A1.** As suggested by the reviewer, we have improved the presentation in the revised version to reduce repetition and include more details about our methods and experiments in response to the questions from reviewers.
>
> > **Q2.1.** I don't quite understand what "step success rate" is as opposed to "success rate" - it's brought up several times as the metric for Mind2Act, but not defined as far as I'm aware.
>
> **A2.1.** In Mind2Web, each step of a task is evaluated separately against the human-annotated ground truth. The step success rate (Step SR) relates to how well the action at each step is performed, whereas the success rate (SR) reflects the success of the entire task, also known as the episode success rate. For example, to complete the task "Show me the reviews for the nearest auto repair business.", the agent has to execute several steps, including typing the search query and clicking through to find the reviews. If all steps in the task are successful, then the task is considered successful. The definition of these metrics has been added in Sec. 4.1.
>
> > **Q2.2.** It doesn't seem to be made clear what exactly the state abstraction entails. In figure 1, the only difference between "raw state" and "clean state" is `<body><h1>…</h1></body>” --> <input id=”where” type=”text”>`. There aren't clear examples in the main text, but intuition about this would be very useful.
>
> **A2.2.** We have reorganized Sec. 3.2 to clarify the state abstraction procedure and included more examples in Appendix C for illustrations. Fig. 1 aims to show that raw states can be very long-winded, which we omitted in  `<h1>…` , while clean observations are much more concise with only task-relevant information, like `<input id="where" type="text">`.

---

> ### Author Response · Authors · 2023-11-17
> **Response to Reviewer z7sh (2/3)**
>
> > **Q2.3.** Not clear how SYNAPSE's history-in-prompt approach is significantly different from Mind2Act's (or lots of other works') to the point of being novel.
>
> **A2.3.** Directly providing trajectories (with multiple states-action pairs) results in very long contexts, as a single webpage state is already very complex. However, current LLMs can only process limited context, and the performance becomes worse as the context becomes longer. As a result, existing computer agents such as RCI and MindAct do not and **CANNOT** utilize **trajectories** as historical information. Instead, their history only consists of **actions**. In comparison, our state abstraction shrinks the length of each state, which is a prerequisite for trajectory-as-exemplar (TaE) prompting.
>
> It is clear that TaE prompting **provides more information** for decision-making. More importantly, TaE prompting uses an interleaved observation-action format, which offers the following advantages over previous prompting schemes.
>
> - **Temporal abstraction**. The interleaved prompt scheme implicitly prompts the LLM to generate multiple actions and query new states only when necessary, as shown in Fig. 2. This significantly improves the success rate of long-horizon tasks in MiniWoB++. In contrast, RCI and MindAct cannot prompt LLMs to generate temporally abstracted actions.
> - **Lower cost and latency** (compared to MindAct). MindAct introduces accumulated error and much higher cost and latency. Specifically, it first filters the top-50 elements with a ranking model. After that, it recursively samples five elements and prompts the LLM to select one from the five until only one is left. As a result, MindAct uses an average of 10x more tokens compared to Synapse in the three test sets of Mind2Web.
> - **Simplified action parsing**. TaE prompting is more effective in prompting LLMs to generate actions in a specific format, which allows easily parsing actions from LLM responses, leading to fewer action parsing errors than other prompting approaches. Additionally, we can set the stop token to "Observation" when querying actions so that the LLM does not generate other information.
> - **Simplified data collection**. Compared to high-level semantic plans or multi-choice questions that are human-crafted in natural language, our approach makes data collection simpler and scalable. The exemplary trajectories can be converted directly from human demonstrations, and the state abstraction prompts can also be inferred from the action with minimal human effort.
>
> In short, our approach is straightforward and very effective, rendering it universally applicable to current agents without much effort.
>
> > **Q2.4.** From both Fig 1 and the text, it's unclear when new trajectory exemplars and example state abstractions are generated - is it per task (that's what I'd think) or per step?
>
> **A2.4.** The trajectory exemplars are used per task. We explained the pipeline of Synapse at the beginning of Sec. 3. Given a task, Synapse retrieves trajectories of similar tasks from memory, which will be fixed during the task execution. We have also added some illustrative trajectories in Appendix C.
>
> > **Q2.5.** Fig 3: might help to have clearer visual distinction of SYNAPSE and the human baseline, especially since the colors aren't always visible. Lines/shading/callout boxes could help.
>
> **A2.5.** Although we agree with the reviewer, it is difficult to adjust the plot. The box plot of Synapse is almost visible because our method is robust and has relatively low variance.

---

> ### Author Response · Authors · 2023-11-17
> **Response to Reviewer z7sh (3/3)**
>
> > **Q3.1.** Though the approach is technically unique and clearly effective, there are some originality concerns in terms of what the paper claims. For example, the paper positions itself against works like Inner Monologues and SayCan by saying that they focus on high-level planning; however, though RCI uses even higher level planning, SYNAPSE is still working with a discrete, abstract action space. Meanwhile, the low-level BC/RL/motion planning+control used for the robotics papers cited are operating in a much more granular, potentially continuous, action space.
>
> **A3.1.** The action spaces we used in MiniWoB++ and Mind2Web align with baseline LLM-powered computer agents [1,2]. The difference we want to highlight is not the granularity of planning or action space. Instead, we propose a prompting scheme that is more effective for action grounding, while Inner Monologues, SayCan, and RCI focused on prompting schemes for better high-level plan generation. Specifically, the robotics methods (Inner Monologues and SayCan) need low-level BC/RL policies to ground high-level plans generated by LLMs into motor control actions because using LLMs for direct robot control is challenging (and unnecessary). The planning granularity of RCI is similar to Inner Monologues and SayCan, but applied in computer control. Therefore, it also requires additional grounding steps to translate high-level plans into executable keyboard-mouse actions. However, the low-level computer actions are highly semantic and can be generated with LLMs, which is quite different from motor control in robotics. However, the suitable prompting scheme here remains unexplored. We have updated this discussion in Sec. 2.
>
> > **Q3.2.** For another example, the paper spends considerable space discussing how it is unique compared to other prompting-based methods in helping the LLM understand the current state. However, MindAct gives trajectory-so-far back to the LLM when prompting for the next step. SYNAPSE gives observation as well, thereby informing the LLM in more detail about the current state, this just isn't a very original idea. It's an important step past MindAct, but not a big one, especially relative to how much it's discussed as a unique contribution.
>
> **A3.2.** While we appreciate the reviewer’s feedback, we respectfully have a different view. We believe our approach is a unique and original contribution to the field due to the reasons discussed in **A2.3**.
>
> > **Q4.** Page 5, last paragraph: Obseravtion is a misspelling
>
> **A4.** Thank you for pointing this out. This typo has been corrected.
>
> > **Q5.** It is inherently interesting that this approach does not need self-correction unlike other prompting-based approaches, but does that offer any significant performance/efficiency gain?
>
> **A5.** Our non-self-correction approach outperforms self-correction baselines, which means that ours has better robustness and lower latency. In fact, Synapse is orthogonal to self-correction methods. We did not add self-correction because it is not our core contribution, and our approach already achieves SOTA performance without it. In contrast, RCI proposed several self-correction modules (task/state/agent grounding), and their average success rate will decrease by around 50% without either one of them, according to their paper [1].
>
> [1] Kim et al. "Language models can solve computer tasks." NeurIPS 2023.
>
> [2] Deng et al. "Mind2Web: Towards a Generalist Agent for the Web." NeurIPS 2023.

---

### Meta-Review · Area_Chair_qLF4 · 2023-12-10

**Metareview:**

This paper presents a prompting method for LLM agents that can perform computer control (eg. issuing terminal calls, navigating through web). The proposed method differs from previous literature in that instead of including action sequences issued so far as part of the prompt, it manages to include the entire trajectory (states and actions) seen so far. To avoid overfilling the conext with state information (eg long HTML pages) the paper introduces a state abstraction method that essentially summarizes raw states into shorter text, which is included as part of the trajectory exemplars in the prompt. The proposed LLM agent also makes use of memory, a vector database that stores task metadata as keys and trajectory exemplars as values, which is used to collect relevant examples of previous solutions into the prompt. All in all, the proposed method significantly outperforms existing methods, and reviewers are unanimous that it deserves to be included in the conference. There are some concerns about the novelty of using trajectory exemplars in the prompt (compared to MindAct), and concerns about existing methods that use a vector database for memory for LLM agents, but there is sufficient empirical advances made here to recommend publication.

**Justification For Why Not Higher Score:**

To the best of my understanding, there is no training or learning occuring in this paper, only prompting. Also, similar ideas behind the proposed prompting have appeared in related literature in the last two years.

**Justification For Why Not Lower Score:**

I don't think the paper deserves to be rejected.

---

### Decision · Program_Chairs · 2024-01-16

Accept (poster)